# SpecTr-GBV: Multi-Draft Block Verification Accelerating Speculative Decoding

## Abstract

Autoregressive language models achieve state-of-the-art performance across a wide range of natural language processing tasks, but suffer from high inference latency due to their sequential decoding nature. Speculative decoding (SD) mitigates this by employing a lightweight draft model to propose candidate tokens, which are selectively verified by a larger target model. While existing methods either adopt multi-draft strategies to increase acceptance rates or block verification techniques to jointly verify multiple tokens, they remain limited by treating these improvements in isolation. In this work, we propose SpecTr-GBV, a novel SD method that unifies multi-draft and greedy block verification (GBV) into a single framework. By formulating the verification step as an optimal transport problem over draft and target token blocks, SpecTr-GBV improves both theoretical efficiency and empirical performance. We theoretically prove that SpecTr-GBV achieves the optimal expected number of accepted tokens for any fixed number of draft sequences, and this bound improves as the number of drafts increases. Empirically, we evaluate SpecTr-GBV across five datasets and four baselines. Our method achieves superior speedup and significantly higher block efficiency while preserving output quality. In addition, we perform comprehensive ablation studies to evaluate the impact of various hyperparameters in the model.

## 1 Introduction

Autoregressive language models achieve state-of-the-art performance across diverse natural language processing tasks but suffer from high latency due to the sequential nature of token generation, where each token requires a full forward pass through the model (Brown et al., 2020; Touvron et al., 2023; Stern et al., 2018). To mitigate the latency challenge, speculative decoding (SD) (Leviathan et al., 2023; Chen et al., 2023) techniques have been developed, employing a smaller draft model to propose candidate tokens that are selectively verified by the target model while preserving distributional fidelity through a sequence of rejection sampling steps.

Standard SD is typically limited to handling a single draft sequence. This constraint leads to only a small number of tokens being accepted in each SD iteration (Sun et al., 2023). To address this limitation, several recent works (Sun et al., 2023; 2024a; Hu et al., 2025) have proposed multi-draft SD strategies. Among them, **SpecTr** (Sun et al., 2023) stands out as a representative method that leverages optimal transport (OT) (Villani et al., 2008) for multi-draft verification. In SpecTr, given a prefix, the draft model is used to generate multiple i.i.d. draft sequences. For each token position (i.e., column across drafts, see Fig. 1a), SpecTr employs OT to compute the optimal coupling between the draft and target distributions. Based on this coupling, it determines which draft token can be accepted, or whether all candidate tokens at that position should be rejected. It can be theoretically shown that as the number of draft sequences increases, the probability of accepting a draft token also increases, thereby allowing to accept more tokens on average (Sun et al., 2023).

SpecTr adopts a position-by-position verification procedure, i.e., the verification iterates over the draft positions sequentially until one of the following conditions is met: either all positions have one accepted token, in which case an extra token is sampled according to the target distribution; or some position has no accepted tokens, in which case all subsequent tokens are discarded, and the token at that position is sampled from a residual distribution. However, Sun et al. (2024b) proved that the position-by-position verification procedure does not yield the optimal expected number of

accepted tokens (see Lemma 1 in Sun et al. (2024b)). To address this, they proposed **greedy block verification (GBV)** for the single-draft setting, which verifies the entire block of draft tokens jointly rather than checking each token independently. Theoretically, GBV achieves the optimal expected number of accepted tokens within a single iteration.

This naturally raises a key question: since both multi-draft and GBV improve the number of accepted tokens, **can we combine the two to further enhance decoding speedup?** In principle, integrating advances from both multi-draft and GBV could yield further gains—this insight forms the basis of our proposed approach. We propose SpecTr with greedy block verification (**SpecTr-GBV**), which extends SpecTr by replacing its position-by-position verification with GBV. Alternatively, it can be viewed as an extension of GBV from the single-draft setting to the multi-draft setting.

The high-level idea of SpecTr-GBV is to generate multiple i.i.d. draft sequences, and formulate the verification procedure as an OT problem between the multiple draft token blocks and the target token block. Theoretically, we show that SpecTr-GBV preserves the fidelity of the output distribution. Moreover, for a fixed number of draft sequences, it achieves the optimal expected number of accepted tokens in each iteration. As the number of draft sequences increases, this optimal expected number also increases accordingly. To the best of our knowledge, our work is the first method unifying multi-draft and block verification strategies within a single framework.

Our main contributions are as follows:

- We propose a novel SD method, SpecTr-GBV, which unifies multi-draft and block verification strategies, thereby achieving longer acceptance lengths and consequently reducing the number of target model calls to improve decoding efficiency.

- We theoretically prove that SpecTr-GBV achieves the optimal expected acceptance length for any fixed number of draft sequences within a single iteration. Moreover, this optimal expected number increases as the number of draft sequences grows.

- We conduct extensive comparisons on five datasets against four baselines. The results show that SpecTr-GBV consistently outperforms standard SD, SpecTr, and GBV in terms of both block efficiency and speedup ratio. In addition, we perform comprehensive ablation studies to evaluate the impact of various hyperparameters in the model.

## 2 RELATED WORK

SD has been widely studied in both single-draft and multi-draft settings. In the single-draft case, most prior work focuses on improving the drafting phase using techniques such as model distillation or document retrieval (Zhou et al., 2023; Liu et al., 2023; He et al., 2023; Fu et al., 2024; Zhang et al., 2023), while relatively fewer efforts target verification. Notably, Sun et al. (2024b;c) improve efficiency by verifying token blocks jointly instead of token-by-token. In the multi-draft setting, recent work explores more advanced verification strategies. Some generate i.i.d. drafts (Sun et al., 2023; Khisti et al., 2024), while others construct tree-structured candidates across steps (Miao et al., 2024; Chen et al., 2024; Yang et al., 2024; Lu et al., 2024). Theoretical studies such as Sun et al. (2024a) and Hu et al. (2025) analyze optimal acceptance under different sampling schemes. These approaches typically refine token-level verification. In contrast to prior work that focuses solely on either multi-draft or block-level verification, our method unifies both by extending block verification to the multi-draft setting, offering stronger theoretical guarantees and improved decoding efficiency.

## 3 PRELIMINARIES

In this section, we provide an overview of SD, SpecTr and GBV.

### 3.1 SPECULATIVE DECODING

Autoregressive sampling in large language models (LLMs) is inherently sequential and often inefficient. SD (Leviathan et al., 2023; Chen et al., 2023) mitigates this inefficiency by offloading token

generation to a smaller draft model, denoted as $\mathcal{M}_s$. Given a prefix $c$, the draft model sequentially generates a candidate sequence of length $L$ as follows:

$$x^{(i)} \sim \mathcal{M}_s(\cdot \mid c, x^{i-1}), \quad i = 1, \ldots, L,$$

where $\mathcal{M}_s(\cdot \mid c, x^{i-1})$ represents the conditional distribution of draft model over the token space, $x^{(i)}$ is the $i$-th generated token, and $x^{i-1} = \{x^{(1)}, \ldots, x^{(i-1)}\}$ denotes the sequence of previously generated tokens. The candidates are then verified in parallel by the target model, denoted as $\mathcal{M}_b$, which provides the true conditional distributions $\{\mathcal{M}_b(\cdot \mid c, x^{i-1})\}_{i=1}^{L+1}$. In the following, we simplify notation by letting $p(\cdot \mid x^{i-1}) = \mathcal{M}_s(\cdot \mid c, x^{i-1})$ and $q(\cdot \mid x^{i-1}) = \mathcal{M}_b(\cdot \mid c, x^{i-1})$. Verification is performed via a sequence of token-level rejection sampling. Each draft token $x^{(i)}$ is accepted with probability:

$$\alpha(x^{(i)}) = \min \left\{ 1, \frac{q(x^{(i)} \mid x^{i-1})}{p(x^{(i)} \mid x^{i-1})} \right\}.$$

If $x^{(i)}$ is the first token to be rejected, all subsequent tokens are discarded, and a replacement is sampled from a residual distribution:

$$p_{\text{res}}(\cdot \mid x^{i-1}) = \text{norm} \left( \max \left\{ q(\cdot \mid x^{i-1}) - p(\cdot \mid x^{i-1}), 0 \right\} \right),$$

where $\text{norm}(\cdot)$ denotes normalization. If all $L$ draft tokens are accepted, an additional token is sampled from the target model: $x^{(L+1)} \sim \mathcal{M}_b(\cdot \mid c, x^L)$. The accepted tokens are appended to the current prefix, and the process is repeated until an end-of-sequence token is generated.

### 3.2 SPECTR

Standard SD relies on a single draft sequence. To improve acceptance rates, SpecTr (Sun et al., 2023) extends SD to the multi-draft setting by generating $K$ i.i.d. draft sequences given a prefix, thereby expanding the candidate space. For each position $i$ (i.e., each column across the drafts, see Fig. 1a), the goal is to accept one valid token from the $K$ candidates $x_1^{(i)}, \ldots, x_K^{(i)}$. SpecTr formulates this verification step as an OT problem. Define $p^{\oplus K}(\cdot \mid x^{i-1})$ as the product distribution over $K$ independent samples from $p(\cdot \mid x^{i-1})$. The goal is to minimize the cost function $C$:

$$\min_{\pi \in \Pi(p^{\oplus K}, q)} C(\pi) = \mathbb{E}_{x_1, \ldots, x_K, y \sim \pi} \left[ \mathbb{I}(y \notin \{x_1, \ldots, x_K\}) \right],$$

where $\mathbb{I}(\cdot)$ is the indicator function, $\pi(x_1, \ldots, x_K, y)$ is a coupling between $p^{\oplus K}$ and $q$, satisfying marginal constraints: $\sum_y \pi = p^{\oplus K}$, $\sum_{x_1, \ldots, x_K} \pi = q$. $\Pi(\cdot, \cdot)$ denotes the set of all valid couplings.

The discrete OT problem above can be solved via linear programming (Cuturi, 2013; Kantorovich, 2006), but its runtime is exponential in $k$ (Sun et al., 2023; Den Hollander, 2012; Pele & Werman, 2009), making it impractical for large vocabularies or draft counts. To address this, Sun et al. (2023) propose K-SEQ, an efficient approximation that iterates over draft tokens at each position and accepts each token with probability:

$$\alpha(x^{(i)}) = \min \left\{ 1, \frac{q(x^{(i)} \mid x^{i-1})}{\rho \cdot p(x^{(i)} \mid x^{i-1})} \right\},$$

where $\rho \in [1, K]$ is a scaling factor determined by solving the equation $1 - (1 - \beta(\rho))^K = \rho \cdot \beta(\rho)$ with $\beta(\rho) = \sum_x \min \left\{ p(x \mid x^{i-1}), \frac{q(x \mid x^{i-1})}{\rho} \right\}$. The computational complexity is given by $O(|\Omega| \log(K))$, with $|\Omega|$ denoting the vocabulary size.

For each position, K-SEQ outputs the first accepted token, and draft sequences matching the accepted token are retained for the next step. If no candidate is accepted, all subsequent tokens are discarded, and a replacement is sampled from a residual distribution:

$$p_{\text{res}}(\cdot \mid x^{i-1}) = \frac{q(\cdot \mid x^{i-1}) - \min \left\{ p(\cdot \mid x^{i-1}), \frac{q(\cdot \mid x^{i-1})}{\rho} \right\} \rho}{1 - \rho \beta(\rho)}.$$

If all positions have accepted tokens, an additional token is sampled from the target model: $x^{(L+1)} \sim \mathcal{M}_b(\cdot \mid c, x^L)$. Theoretically, it can be shown that as $K$ increases, the likelihood of accepting a draft token also increases, which in turn leads to further speedups.

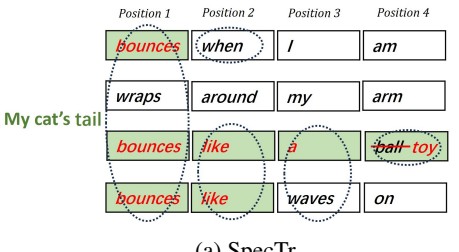
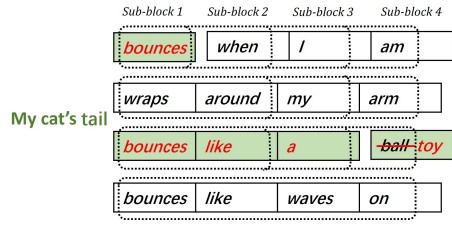

|  (a) SpecTr  |  (b) SpecTr-GBV  |

Figure 1: Comparison between SpecTr and SpecTr-GBV. Given multiple i.i.d. draft sequences, SpecTr performs position-by-position verification: at each position, it accepts one token (e.g., *bounces*, then *like*) and retains only the sequences consistent with accepted tokens. If no token is accepted, a new token is sampled from the residual distribution (e.g., replacing *ball* with *toy*). SpecTr-GBV verifies token sub-blocks across all draft sequences. It starts by jointly verifying sub-blocks with the first position (e.g., *bounces, bounces when, ..., bounces when I am*) and select the longest accepted sub-block (e.g., *bounces*). Verification continues from the next position in the next sequence (e.g., *wraps around, ..., wraps around my arm*). If no sub-block is accepted, we proceed to the next sequence, progressively selecting the longest accepted sub-block (e.g., *bounces like a*). A residual token is then sampled after the longest accepted sub-block (e.g., replacing *ball* with *toy*).

### 3.3 GREEDY BLOCK VERIFICATION

Position-by-position verification terminates when all candidates at a given position are rejected. However, Sun et al. (2024c;b) point out that such a verification strategy does not achieve the optimal expected number of accepted tokens within a single iteration. To address this, they propose GBV in *the single-draft setting*. Unlike standard methods that verify each token independently, GBV evaluates each sub-block $x^i = \{x^{(1)}, \ldots, x^{(i)}\}$ as a whole and accepts it with probability:

$$\alpha(x^i) = \frac{\sum_x \max\left\{\nu_i q(x \mid x^i) - p(x \mid x^i), 0\right\}}{\sum_x \max\left\{p(x \mid x^i) - \nu_i q(x \mid x^i), 0\right\}}, \quad i = 1, \ldots, L-1,$$

where $\nu_i = \nu_{i-1} \frac{q(x^{(i)}|x^{i-1})}{p(x^{(i)}|x^{i-1})}$, $\nu_0 = 1$, $\alpha(x^L) = \nu_L$. Here, $\nu_i$ represents the probability that sub-block $x^i$ is retained in the final output.

The final accepted block is the longest accepted sub-block $x^\tau$ from the above process. If $\tau = L$, an extra token is drawn from the target model: $x^{(L+1)} \sim \mathcal{M}_b(\cdot \mid c, x^L)$. If $\tau < L$, an extra token is sampled from a residual distribution:

$$p_{\text{res}}(\cdot \mid x^\tau) = \text{norm}\left(\max\left\{\nu_\tau q(\cdot \mid x^\tau) - p(\cdot \mid x^\tau), 0\right\}\right),$$

which ensures distributional consistency within a single iteration. To maintain the same output distribution across multiple SD iterations, the target distribution is modified at all unaccepted positions before the next iteration as follows:

$$q(\cdot \mid x^i) = \text{norm}\left(\max\left\{\nu_i q(\cdot \mid x^i) - p(\cdot \mid x^i), 0\right\}\right), \quad \forall \tau + 1 \leq i \leq L-1.$$

Theoretically, GBV achieves the highest expected number of accepted tokens within a single SD iteration among all valid verification algorithms in the single draft setting.

## 4 METHODOLOGY

In this section, we introduce our proposed SpecTr-GBV. We begin by formulating the verification as an optimization problem that aims to maximize the expected acceptance length. We then present the detailed algorithmic procedure.

### 4.1 MAXIMUM ACCEPTANCE LENGTH WITH I.I.D. DRAFT SEQUENCES

To further improve acceptance length, SpecTr-GBV combines SpecTr with GBV. Given a prefix $c$, we generate $K$ i.i.d. draft sequences of length $L$ from the draft model: $X^L = \{x_1^L, x_2^L, \ldots, x_K^L\}$.

We formulate the verification procedure as an OT problem between the multiple draft token blocks $X^L$ and the target token block $y^L$.

**Definition 4.1.** [Acceptance Length] For the $K$ i.i.d. draft sequences $X^L$ and a target sequence $y^L$, define the *acceptance length* as

$$\tau = \max \left\{ i \in \{0, \ldots, L\} \ : \ \exists k \in \{1, \ldots, K\} \text{ such that } x_k^i = y^i \right\}.$$

Our goal is to maximize the expected acceptance length, or equivalently minimize the cost function under the physical constraint of i.i.d. draft generation:

$$\min_{\pi \in \Pi_{\text{CIC}}(p^{\oplus K}, q)} C(\pi) = L - \mathbb{E}_{X^L, y^L \sim \pi} [\tau], \tag{1}$$

where $\Pi_{\text{CIC}}(p^{\oplus K}, q)$ denotes the set of *Conditionally Independent Couplings* (CIC). Unlike general couplings, a coupling $\pi \in \Pi_{\text{CIC}}$ requires that the draft sequences remain independent conditioned on the target sequence $y^L$, i.e., $P_\pi(X^L \mid y^L) = \prod_{k=1}^K P_\pi(x_k^L \mid y^L)$. This structural constraint is necessary to align the theoretical formulation with the physically realizable verification schemes where drafts are generated i.i.d..

However, solving the optimization problem above presents several challenges. On the one hand, the joint distributions $q(x^i)$ and $p(x^i)$ for all sub-blocks are not accessible; on the other hand, no closed-form solution exists for the OT problem when $K > 1$ and solving it via linear programming incurs prohibitive computational overhead. To address these issues, we propose the SpecTr-GBV algorithm. Using this algorithm, the expected acceptance length $\mathbb{E}[\tau]$ strictly attains the theoretical value of maximum expected acceptance length $\max_{\pi \in \Pi_{\text{CIC}}(p^{\oplus K}, q)} \mathbb{E}_{X^L, y^L \sim \pi} [\tau]$, while circumventing the aforementioned challenges and efficiently solving the OT problem with provable guarantees.

## 4.2 SpecTr-GBV

In this section, we present the SpecTr-GBV algorithm, which achieves the maximum expected acceptance length stated in Theorem 5.3. The details of the algorithm are provided in Algorithm 1, and its illustration is shown in Fig. 1b. The core equations governing the verification procedure are:

$$h_{ik} = \frac{\sum_x q(x^i, x) \left(1 - \min\left\{\frac{p(x^i, x)}{q(x^i, x)}, 1\right\}\right)^K - q(x^i)\left(1 - \min\left\{\frac{p(x^i)}{q(x^i)}, 1\right\}\right)^K}{1 - (1 - p(x^i))^K - q(x^i) + \sum_x q(x^i, x)\left(1 - \min\left\{\frac{p(x^i, x)}{q(x^i, x)}, 1\right\}\right)^K}, \tag{2}$$

$$h_{Lk} = \frac{q(x^L)\left[1 - \left(1 - \min\left\{\frac{p(x^L)}{q(x^L)}, 1\right\}\right)^K\right]}{1 - (1 - p(x^L))^K}, \tag{3}$$

$$p_{\text{res}}(x \mid x^\tau) = \frac{q(x^\tau, x)\left(1 - \min\left\{\frac{p(x^\tau, x)}{q(x^\tau, x)}, 1\right\}\right)^K}{\sum_x q(x^\tau, x)\left(1 - \min\left\{\frac{p(x^\tau, x)}{q(x^\tau, x)}, 1\right\}\right)^K}. \tag{4}$$

SpecTr-GBV sequentially iterates over the $K$ i.i.d. draft sequences. The verification for each sequence $x_k^L$ begins at the sub-block $x_k^{\tau+1}$, where $\tau$ denotes the length of the current longest accepted sub-block. Each token sub-block $x_k^i$, for $i = \tau + 1, \ldots, L$, is accepted with probability $h_{ik}$. When a token block is rejected, it is recorded in the set $H$. To avoid redundant sampling, if a token sub-block $x_k^i$ scheduled for verification already exists in $H$, the sampling step for that block is skipped, and the algorithm immediately proceeds to verify the next sub-block $x_k^{i+1}$. After completing verification for all sub-blocks in $x_k^L$, if the entire token block $x_k^L$ is accepted, the sequence-level iteration terminates early; otherwise, SpecTr-GBV proceeds to verify the next draft sequence $x_{k+1}^L$. The final accepted block is the longest accepted sub-block obtained through this process. If the final accepted block equals the entire token block $t^L$, an additional token is sampled directly from the target model: $y \sim \mathcal{M}_b(\cdot \mid c, t^L)$. If the length of the final accepted block $\tau$ satisfies $\tau < L$, an

---

**Algorithm 1** SpecTR-GBV

---

1: **Require:** draft sequences $x_1^L, x_2^L, \ldots, x_K^L$; the conditional distributions of tokens from the draft and target models $p(\cdot \mid x_k^{i-1})$ for $i = 1, \ldots, L$, and $q(\cdot \mid x_k^{i-1})$ for $i = 1, \ldots, L+1, \forall k = 1, \ldots, K$.
2: **Ensure:** output tokens $t, y$.
3: Initialize draft sequence set $S = \{x_k^L \mid k = 1, \ldots, K\}$, acceptance length $\tau = 0$, accepted token sub-block $t$, unaccepted token sub-block set $H = \emptyset$, accepted draft sequence index $f = 0$.
4: **for** each $x_k^L \in S$ **do**
5:      **for** $i = \tau + 1, \ldots, L-1$ **do**
6:          Compute $h_{ik}$ as Eq. (2); sample $\eta \sim U(0, 1)$
7:          **if** $\eta < h_{ik}$ and $x_k^i \notin H$ **then**                ▷ verify sub-block.
8:              $\tau = i, f = k, t = x_k^i$
9:          **else**                           ▷ record unaccepted sub-block.
10:              $H \leftarrow H \cup \{x_k^i\}$
11:          **end if**
12:      **end for**
13:      Compute $h_{Lk}$ as Eq. (3); sample $\eta \sim U(0, 1)$
14:      **if** $\eta < h_{Lk}$ and $x_k^L \notin H$ **then**               ▷ verify entire block.
15:          $\tau = L, t = x_k^L$; sample $y \sim q(\cdot \mid x_k^L)$; **break**
16:      **else**                            ▷ record unaccepted block.
17:          $H \leftarrow H \cup \{x_k^L\}$
18:      **end if**
19: **end for**
20: **if** $\tau < L$ **then**                           ▷ residual sampling.
21:      Sample $y \sim p_{\text{res}}(\cdot \mid x_f^\tau)$ as Eq. (4)
22: **end if**
23: **return** $t, y$

---

extra token is sampled from a residual distribution as defined in Eq. (4). For each selection step for $x_j^i$, SpecTr-GBV exhibits a complexity of $O(|\Omega|)$, equivalent to GBV and more efficient than the $O(|\Omega| \log(K))$ required by SpecTr.

Similar to GBV, in order to maintain distributional consistency across multiple SD iterations, SpecTr-GBV includes a distribution modification step before the next SD iteration begins. Specifically, the next $L - \tau - 1$ tokens need to be sampled from a modified distribution, denoted as $x^{L-\tau-1} \sim q_{\text{new}}(\cdot)$, which differs from the original target distribution $q$. The computation of this modified distribution is shown in Algorithm 2.

---

**Algorithm 2** Distribution Modification

---

**Require:** $p, q, L, t^\tau, y$
**Ensure:** $q_{\text{new}}$
1: For $i \leq L - \tau - 1$:

$$q_{\text{new}}(x^{(i)}) = \frac{q\left(t^\tau, y, x^{i-1}, x^{(i)}\right) \left(1 - \min\left\{\frac{p\left(t^\tau, y, x^{i-1}, x^{(i)}\right)}{q\left(t^\tau, y, x^{i-1}, x^{(i)}\right)}, 1\right\}\right)^K}{\sum_{x'} q\left(t^\tau, y, x^{i-1}, x'\right) \left(1 - \min\left\{\frac{p\left(t^\tau, y, x^{i-1}, x'\right)}{q\left(t^\tau, y, x^{i-1}, x'\right)}, 1\right\}\right)^K}.$$

2: For $i > L - \tau - 1$:

$$q_{\text{new}}(x^{(i)}) = q(x^{(i)}).$$

3: **return** $q_{\text{new}}$

---

**Theorem 4.2** (Distribution Preservation). *SpecTr-GBV preserves the target model distribution. Specifically, for any prefix $c$, given conditional distributions $p$ and $q$ from the draft and target models, draft length $L$, and draft sequences $X^L \sim q^{\oplus K}(\cdot)$, the output $O = (t^\tau, y, x^{L-\tau-1})$ satisfies:*

$$O \sim \textit{SpecTr-GBV}(c, p, q, X^L) \implies O \sim q(\cdot),$$

---

**Algorithm 3** Speculative Decoding with SpecTr-GBV

---

**Require:** prefix $c$, target model distribution $q$, draft model distribution $p$, draft length $L$
**Ensure:** decoded sequence

---

1: **while** End of Sequence $\notin (t^\tau, y)$ **do**
2:      Sample $x_1^L, \ldots, x_K^L \sim p(\cdot)$ i.i.d.                 ▷ obtain draft block.
3:      Compute $q(\cdot \mid x_k^{i-1})$ for $i = 1, \ldots, L+1$ , $k = 1, \ldots, K$ in parallel.     ▷ parallel scoring.
4:      $(t^\tau, y) \leftarrow \text{VERIFY}\left(X^L, \{q(\cdot \mid x_k^{i-1})\}, \{p(\cdot \mid x_k^{i-1})\}\right)$.         ▷ draft verification.
5:      $c \leftarrow c, t^\tau, y$.
6:      $q \leftarrow \text{DistributionModification}(q, p, L, t^\tau, y)$.            ▷ modify distribution.
7: **end while**

---

*where $t^\tau$ is the accepted tokens of length $\tau \in [0, L]$, $y$ is the correction token sampled from the residual distribution $p_{res}(\cdot \mid t^\tau)$ as defined in Eq. (4), and $x^{L-\tau-1}$ is the remaining sub-block sampled from the modified distribution, i.e., $x^{L-\tau-1} \sim q_{new}(\cdot)$.*

By combining Algorithm 1 and Algorithm 2, we arrive at the final complete procedure, Algorithm 3, which functions as a valid SD method. Compared to standard SD methods, Algorithm 3 additionally incorporates a distribution modification step to $\mathcal{M}_b$ before the next SD iteration begins. The proofs of Algorithm 1, Algorithm 2, Algorithm 3, and Theorem 4.2 are provided in Appendix A.

## 5 THEORETICAL ANALYSIS

In this section, we present the theoretical guarantees of SpecTr-GBV. First, we establish the upper bound on the expected acceptance length and characterize its associated properties.

**Lemma 5.1** (Upper Bound on Expected Acceptance Length). *For any coupling $\pi \in \Pi_{CIC}(p^{\oplus K}, q)$, the expected acceptance length is upper bounded by:*

$$\mathbb{E}_{X^L, y^L \sim \pi}[\tau] \leq \sum_{\tau=1}^{L} \sum_{x^\tau} q(x^\tau) \left[ 1 - \left( 1 - \min\left\{ \frac{p(x^\tau)}{q(x^\tau)}, 1 \right\} \right)^K \right]. \tag{5}$$

*Remark* 5.2. The upper bound on the expected acceptance length in Eq. (5) has following properties:

- **Monotonicity**: Let $\text{Bound}(K)$ denote the upper bound value in Eq. (5) for $K$ draft sequences. For any positive integers $K_1$ and $K_2$ satisfying $K_1 > K_2$, the bound strictly increases: $\text{Bound}(K_1) > \text{Bound}(K_2)$.

- **Convergence:** The upper bound converges to the maximum acceptance length $L$ as the number of drafts $K$ tends to infinity: $\lim_{K \to \infty} \text{Bound}(K) = L$.

- **Consistency:** When $K = 1$, the bound reduces to the single-draft setting, which corresponds exactly to GBV: $\text{Bound}(1) = \sum_{\tau=1}^{L} \sum_{x^\tau} \min\{p(x^\tau), q(x^\tau)\}$.

Lemma 5.1 and Remark 5.2 characterize the intrinsic performance limits determined by the marginal distributions and show how multiple i.i.d. drafts strictly improve the upper bound on the expected acceptance length as $K$ increases. This result provides the theoretical foundation for SpecTr-GBV. As stated in the following theorem, the design of SpecTr-GBV ensures that it achieves the maximum expected acceptance length established in Lemma 5.1.

**Theorem 5.3** (SpecTr-GBV Acceptance Length). *The SpecTr-GBV achieves the upper bound of the expected acceptance length:*

$$\mathbb{E}_{X^L, y^L \sim \pi^*}[\tau] = \sum_{\tau=1}^{L} \sum_{x^\tau} q(x^\tau) \left[ 1 - \left( 1 - \min\left\{ \frac{p(x^\tau)}{q(x^\tau)}, 1 \right\} \right)^K \right], \tag{6}$$

*where $\pi^*$ denotes the optimal coupling between $p^{\oplus K}(X^L)$ and $q(y^L)$ that minimizes the cost function in Eq. (1).*

The proofs of Lemma 5.1 and Theorem 5.3 are provided in Appendix B.

## 6 EXPERIMENTS

In this section, we evaluate SpecTr-GBV against four baselines across five benchmark datasets with different combinations of draft and target models. In addition, we perform extensive ablation studies on the draft length $L$, the draft number $K$, and the temperature $T$.

### 6.1 BASELINES AND DATASETS

For baselines, we compare SpecTr-GBV with (1) **Autoregressive Decoding (AR)**, (2) **Speculative Decoding (SD)** (Leviathan et al., 2023; Chen et al., 2023), (3) **SpecTr** (Sun et al., 2023), (4) **Greedy Block Verification (GBV)** (Sun et al., 2024b). For datasets, we evaluate on five diverse tasks: (1) python programming problems (**HumanEval**) (Chen et al., 2021), (2) grade school math problems (**GSM8K**) (Cobbe et al., 2021), (3) multilingual grade school math problems (**MGSM**) (Shi et al., 2022), (4) language modeling with One-Billion Word Benchmark (**LM1B**) (Chelba et al., 2013), (5) Stanford instruction-following dataset (**Alpaca**) (Taori et al., 2023).

### 6.2 METRICS

We report comparison results across two key metrics for all methods. (1) **Block Efficiency (BE)** (Leviathan et al., 2023): the average number of decoded tokens per serial call, defined as: BE = Total number of decoded tokens/Number of serial calls to $\mathcal{M}_b$. (2) **Speedup Ratio (SR)** (Leviathan et al., 2023): the ratio of the wall-clock time of baseline AR to that of the proposed method: SR = $T_{\text{autoregressive}}/T_{\text{proposed}}$.

### 6.3 MAIN RESULTS

We report results on three major LLM families: `DeepSeek` (Guo et al., 2024), `CodeLlama` (Roziere et al., 2023), and `Vicuna` (Chiang et al., 2023). Results for `DeepSeek` are shown in Table 1, while those for `CodeLlama` and `Vicuna` are provided in Appendix C. In Table 1, we report the mean and standard deviation of each metric with different random seeds on $1,000$ test prompts and across multiple datasets under two settings: draft length $L = 12$, temperature $T = 0.4$, draft number $K = 3$, `DeepSeek-33B` as the target, `DeepSeek-1.3B` as the draft, and $L = 8$, $T = 0.4$, $K = 3$, `DeepSeek-6.7B` as the target, `DeepSeek-1.3B` as the draft. In the `33B-1.3B` case, SpecTr-GBV achieves a $12.4\%$ improvement in average BE and a $29.3\%$ gain in average SR compared with SD, a $2.3\%$ improvement in average BE and an $8.2\%$ gain in average SR compared with SpecTr, a $9.7\%$ improvement in average BE and a $27.0\%$ gain in average SR compared with GBV respectively. In the `6.7B-1.3B` case, SpecTr-GBV achieves an $11.6\%$ improvement in average BE and a $14.3\%$ gain in average SR compared with SD, a $2.2\%$ improvement in average BE and an $8.1\%$ gain in average SR compared with SpecTr, a $9.1\%$ improvement in average BE and a $13.2\%$ gain in average SR compared with GBV respectively.

The SR is smaller than the BE, which aligns with our expectations. This discrepancy primarily arises from two factors: First, the draft model also requires non-negligible computation time to generate tokens, which cannot be ignored in practice. Second, during verification, the target model executes batch inference, which leads to slightly higher latency for individual requests compared to single-batch inference.

### 6.4 ABLATION STUDIES

We analyze the sensitivity of draft length $L$, draft number $K$, and temperature $T$ in the `33B-1.3B` setting. The experiments are conducted on the HumanEval dataset with five different random seeds. The results in the `6.7B-1.3B` setting are shown in Appendix C. **(1) Effect of draft length $L$:** We compare SpecTr-GBV against baselines under varying draft lengths $L = 12, 16, 20, 24$ with temperature $T = 0.4$ and draft number $K = 3$. As shown in Table 2, SpecTr-GBV consistently outperforms all baselines across different settings. More importantly, as $L$ increases, BE steadily improves, while SR first increases and then declines. This is because longer draft lengths incur higher computational overhead during the draft phase, which eventually outweighs the benefits of reduced target model calls. **(2) Effect of draft number $K$:** We compare the acceptance rates of SpecTr-GBV and SpecTr under different numbers of draft sequences $K = 1, 3, 5, 7$ with draft lengths $L = 12$ and 16. As

Table 1: Performance comparison of SpecTr-GBV with baselines. The results demonstrate that SpecTr-GBV consistently outperforms all baselines across five datasets, regardless of whether `DeepSeek-33B` or `DeepSeek-6.7B` is used as the target model.

| Setting | Dataset | Metric | AR | SD | SpecTr | GBV | SpecTr-GBV |
|---|---|---|---|---|---|---|---|
| DeepSeek-33B DeepSeek-1.3B L=12 T=0.4 K=3 | HumanEval | BE | 1 | $9.53 \pm 1.53$ | $10.25 \pm 1.37$ | $9.60 \pm 1.52$ | $\mathbf{10.45 \pm 1.44}$ |
| | | SR | 1 | $2.02 \pm 0.32$ | $2.21 \pm 0.30$ | $2.03 \pm 0.32$ | $\mathbf{2.50 \pm 0.39}$ |
| | GSM8K | BE | 1 | $6.83 \pm 1.66$ | $7.25 \pm 1.31$ | $6.89 \pm 1.44$ | $\mathbf{7.65 \pm 1.54}$ |
| | | SR | 1 | $1.57 \pm 0.42$ | $2.26 \pm 0.44$ | $1.57 \pm 0.37$ | $\mathbf{2.46 \pm 0.58}$ |
| | MGSM | BE | 1 | $7.18 \pm 1.85$ | $8.10 \pm 1.86$ | $7.37 \pm 1.88$ | $\mathbf{8.31 \pm 1.81}$ |
| | | SR | 1 | $1.55 \pm 0.45$ | $1.97 \pm 0.72$ | $1.60 \pm 0.47$ | $\mathbf{2.12 \pm 0.76}$ |
| | LM1B | BE | 1 | $8.17 \pm 2.76$ | $8.93 \pm 2.45$ | $8.42 \pm 2.66$ | $\mathbf{8.94 \pm 2.51}$ |
| | | SR | 1 | $1.71 \pm 0.57$ | $1.82 \pm 0.47$ | $1.75 \pm 0.54$ | $\mathbf{1.90 \pm 0.55}$ |
| | Alpaca | BE | 1 | $6.51 \pm 2.28$ | $7.42 \pm 2.18$ | $6.89 \pm 2.27$ | $\mathbf{7.58 \pm 2.22}$ |
| | | SR | 1 | $1.36 \pm 0.47$ | $1.52 \pm 0.42$ | $1.41 \pm 0.54$ | $\mathbf{1.60 \pm 0.48}$ |
| | Average | BE | 1 | 7.64 | 8.39 | 7.83 | **8.59** |
| | | SR | 1 | 1.64 | 1.96 | 1.67 | **2.12** |
| DeepSeek-6.7B DeepSeek-1.3B L=8 T=0.4 K=3 | HumanEval | BE | 1 | $7.06 \pm 0.91$ | $7.53 \pm 0.81$ | $7.06 \pm 0.92$ | $\mathbf{7.70 \pm 0.72}$ |
| | | SR | 1 | $1.19 \pm 0.15$ | $1.15 \pm 0.18$ | $1.19 \pm 0.15$ | $\mathbf{1.28 \pm 0.12}$ |
| | GSM8K | BE | 1 | $5.78 \pm 0.55$ | $6.55 \pm 0.58$ | $5.99 \pm 0.65$ | $\mathbf{6.68 \pm 0.72}$ |
| | | SR | 1 | $1.02 \pm 0.10$ | $1.28 \pm 0.11$ | $1.04 \pm 0.12$ | $\mathbf{1.42 \pm 0.17}$ |
| | MGSM | BE | 1 | $5.84 \pm 1.00$ | $6.38 \pm 1.01$ | $5.99 \pm 1.01$ | $\mathbf{6.60 \pm 1.11}$ |
| | | SR | 1 | $1.01 \pm 0.18$ | $1.06 \pm 0.26$ | $1.02 \pm 0.20$ | $\mathbf{1.14 \pm 0.31}$ |
| | LM1B | BE | 1 | $6.17 \pm 1.78$ | $6.86 \pm 1.66$ | $6.42 \pm 1.73$ | $\mathbf{7.01 \pm 1.66}$ |
| | | SR | 1 | $1.02 \pm 0.29$ | $1.04 \pm 0.22$ | $1.03 \pm 0.27$ | $\mathbf{1.12 \pm 0.27}$ |
| | Alpaca | BE | 1 | $5.81 \pm 1.28$ | $6.12 \pm 1.18$ | $5.89 \pm 1.27$ | $\mathbf{6.23 \pm 0.22}$ |
| | | SR | 1 | $1.01 \pm 0.17$ | $1.00 \pm 0.42$ | $1.00 \pm 0.14$ | $\mathbf{1.03 \pm 0.18}$ |
| | Average | BE | 1 | 6.13 | 6.69 | 6.27 | **6.84** |
| | | SR | 1 | 1.05 | 1.11 | 1.06 | **1.20** |

Table 2: Ablation results of SpecTr-GBV under different draft lengths $L$.

| $L$ | AR | | SD | | SpecTr | | GBV | | SpecTr-GBV | |
|---|---|---|---|---|---|---|---|---|---|---|
| | BE | SR | BE | SR | BE | SR | BE | SR | BE | SR |
| 12 | 1 | 1 | 9.53 | 2.02 | 10.25 | 2.21 | 9.60 | 2.03 | **10.45** | **2.50** |
| 16 | 1 | 1 | 11.36 | 1.93 | 12.56 | 2.56 | 11.37 | 1.94 | **12.63** | **2.52** |
| 20 | 1 | 1 | 13.10 | 1.88 | 13.64 | 2.55 | 13.19 | 1.86 | **14.58** | **2.87** |
| 24 | 1 | 1 | 14.63 | 1.80 | 16.22 | 2.33 | 15.08 | 1.85 | **16.52** | **2.68** |

shown in Fig. 2a, and consistent with our theoretical analysis, the acceptance rates of both methods increase as $K$ grows. Moreover, SpecTr-GBV consistently achieves higher acceptance rates than SpecTr across all settings. **(3) Effect of temperature** $T$: We evaluate the BE and SR of SpecTr-GBV under different temperatures $T = 0.1, 0.4, 0.7$ with draft lengths $L = 12$ and $16$. As shown in Fig. 2b, both metrics exhibit minimal variation across temperatures, demonstrating the robustness of SpecTr-GBV to changes in temperature.

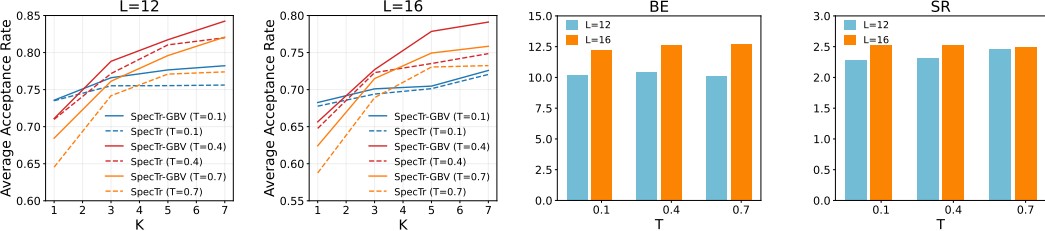

(a) Acceptance rate comparison between SpecTr-GBV and SpecTr with varying $K$ ($L = 12$ and $16$).

(b) BE and SR performance of SpecTr-GBV with varying temperature $T$ ($L = 12$ and $16$).

Figure 2: Ablation results of SpecTr-GBV under different draft number (a) and temperature (b).

## 7 CONCLUSIONS

In this work, we propose SpecTr-GBV, a novel speculative decoding framework that unifies multi-draft generation with greedy block verification. Our method formulates the verification process as an optimal transport problem between draft and target token blocks, enabling more effective token acceptance within each iteration. Theoretically, we show that SpecTr-GBV achieves the optimal expected number of accepted tokens for a fixed number of draft sequences, and this optimal bound increases with more drafts. Empirically, extensive experiments across five datasets and multiple baselines demonstrate that SpecTr-GBV significantly improves both block efficiency and decoding speed while preserving output distribution fidelity.

## ETHICS STATEMENT

Our work adheres to the ICLR Code of Ethics. This study did not involve human subjects or animal experimentation. All datasets were used in compliance with relevant guidelines, ensuring that no privacy regulations were violated. We took care to minimize potential biases and avoid discriminatory outcomes throughout the research process. No personally identifiable information was included, and no experiments were conducted that could raise privacy or security concerns. We remain committed to transparency, integrity, and ethical responsibility in all aspects of this research.

## REPRODUCIBILITY STATEMENT

We have taken extensive measures to ensure the reproducibility of our research. The complete source code for the proposed method is provided in the supplementary materials to facilitate replication and verification. Detailed descriptions of the experimental setup, baselines, and evaluation metrics are included in the Experiments section, along with a full description of the algorithm to support faithful reproduction. All datasets used are publicly available, ensuring consistent and verifiable evaluation results. We believe these measures will enable other researchers to reproduce our findings and further advance the field.

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

# A    PROOF OF THEOREM 4.2

We begin with the following lemma, which characterizes the probability that SpecTr-GBV accepts each sub-block.

**Lemma A.1** (Probability of Accepting Sub-block). *For all $i \in [1, L]$ and $X^L$, in SpecTr-GBV, we have*

$$P\big(X^L \text{ has } x^i, \tau^{x^i} \geq i\big) = q(x^i) \left[ 1 - \left( 1 - \min\left\{ \frac{p(x^i)}{q(x^i)}, 1 \right\} \right)^K \right].$$

*where $\tau^{x^i}$ denotes the acceptance length of token block $x^i$.*

*Proof.* We prove this by induction in the backward direction. When $i = L$, this holds due to the maximal coupling step for the entire block $x^L$ in Algorithm 1 since

$$P\big(X^L \text{ has } x^L, \tau^{x^L} \geq L\big) = P\big(X^L \text{ has } x^L\big) P\left( \tau^{x^L} = L \,\Big|\, X^L \text{ has } x^L \right)$$

$$= \left[ 1 - \big(1 - p(x^L)\big)^K \right] \cdot \frac{q(x^L) \left[ 1 - \left( 1 - \min\left\{ \frac{p(x^L)}{q(x^L)}, 1 \right\} \right)^K \right]}{1 - (1 - p(x^L))^K}$$

$$= q(x^L) \left[ 1 - \left( 1 - \min\left\{ \frac{p(x^L)}{q(x^L)}, 1 \right\} \right)^K \right].$$

where $P\left( \tau^{x^L} = L \,\Big|\, X^L \text{ has } x^L \right)$ denotes $h_{Lk}$ in Eq. (3).

Suppose the equation holds for $i \geq i_0$. For $i = i_0 - 1$, we have

$$P\big(X^L \text{ has } x^{i_0-1}, \tau^{x^{i_0-1}} \geq i_0 - 1\big) = P\big(X^L \text{ has } x^{i_0-1}, \tau^{x^{i_0-1}} \geq i_0\big)$$
$$+ P\big(X^L \text{ has } x^{i_0-1}, \tau^{x^{i_0-1}} = i_0 - 1\big).$$

For the first term, by the induction assumption:

$$P\big(X^L \text{ has } x^{i_0-1}, \tau^{x^{i_0-1}} \geq i_0\big) = \sum_x P\big(X^L \text{ has } (x^{i_0-1}, x), \tau^{(x^{i_0-1}, x)} \geq i_0\big)$$

$$= \sum_x q(x^{i_0-1}, x) \left[ 1 - \left( 1 - \min\left\{ \frac{p(x^{i_0-1}, x)}{q(x^{i_0-1}, x)}, 1 \right\} \right)^K \right]. \quad (7)$$

For the second term, define the acceptance probability:

$$h \triangleq P\big(\text{accept } x^{i_0-1} \,\big|\, X^L \text{ has } x^{i_0-1}\big).$$

Then:

$$P\big(X^L \text{ has } x^{i_0-1}, \tau^{x^{i_0-1}} = i_0 - 1\big) = P\big(X^L \text{ has } x^{i_0-1}, \tau < i_0\big) P\big(\text{accept } x^{i_0-1} \,\big|\, X^L \text{ has } x^{i_0-1}\big)$$
$$= \big(P\big(X^L \text{ has } x^{i_0-1}\big) - P\big(X^L \text{ has } x^{i_0-1}, \tau \geq i_0\big)\big) h. \quad (8)$$

The marginal probability is:

$$P\big(X^L \text{ has } x^{i_0-1}\big) = 1 - \big(1 - p\big(x^{i_0-1}\big)\big)^K. \quad (9)$$

Substituting Eq. (9) into Eq. (8):

$$P\big(X^L \text{ has } x^{i_0-1}, \tau^{x^{i_0-1}} = i_0 - 1\big) = \left( \left[ 1 - \big(1 - p(x^{i_0-1})\big)^K \right] - \sum_x q(x^{i_0-1}, x) \cdot \right.$$

$$\left. \left[ 1 - \left( 1 - \min\left\{ \frac{p(x^{i_0-1}, x)}{q(x^{i_0-1}, x)}, 1 \right\} \right)^K \right] \right) h. \quad (10)$$

Combining terms Eqs. (7) and (10) yields:

$$P\big(X^L \text{ has } x^{i_0-1},\ \tau^{x^{i_0-1}} \geq i_0 - 1\big)$$

$$= P\big(X^L \text{ has } x^{i_0-1},\ \tau^{x^{i_0-1}} \geq i_0\big) + P\big(X^L \text{ has } x^{i_0-1},\ \tau^{x^{i_0-1}} = i_0 - 1\big)$$

$$= \sum_x q(x^{i_0-1}, x)\Big[1 - \big(1 - \min\{\tfrac{p(x^{i_0-1},x)}{q(x^{i_0-1},x)}, 1\}\big)^K\Big] +$$

$$\Big(\big[1 - (1 - p(x^{i_0-1}))^K\big] - \sum_x q(x^{i_0-1}, x)\Big[1 - \big(1 - \min\{\tfrac{p(x^{i_0-1},x)}{q(x^{i_0-1},x)}, 1\}\big)^K\Big]\Big) \cdot h. \qquad (11)$$

As in Eq. (2), $h_{(i_0-1)k}$ equals to

$$h = \frac{\sum_x q(x^{i_0-1}, x)\left(1 - \min\left\{\frac{p(x^{i_0-1},x)}{q(x^{i_0-1},x)}, 1\right\}\right)^K - q(x^{i_0-1})\left(1 - \min\left\{\frac{p(x^{i_0-1})}{q(x^{i_0-1})}, 1\right\}\right)^K}{1 - (1 - p(x^{i_0-1}))^K - q(x^{i_0-1}) + \sum_x q(x^{i_0-1}, x)\left(1 - \min\left\{\frac{p(x^{i_0-1},x)}{q(x^{i_0-1},x)}, 1\right\}\right)^K},$$

By the verification mechanism, substituting $h$ into Eq. (11) yields:

$$P\big(X^L \text{ has } x^{i_0-1},\ \tau^{x^{i_0-1}} \geq i_0 - 1\big) = q(x^{i_0-1})\left[1 - \left(1 - \min\left\{\frac{p(x^{i_0-1})}{q(x^{i_0-1})}, 1\right\}\right)^K\right], \qquad (12)$$

which completes the proof. □

Here, we continue to prove the Theorem 4.2 which demonstrates that complete procedure of SpecTr-GBV, as described in Algorithm 3, preserves the distribution consistency of target model distribution. That is, for $\forall i \leq L$ and sub-block $x^i$, we have $P\big(O^i = x^i\big) = q(x^i)$.

*Proof.* We prove the claim by induction. Note that the corollary $P\big(O^i = x^i\big) = p(x^i)$ holds for $i = 0$, which is the trivial case and both sides are equal to 1. Suppose the claim holds for $i \leq i_0$. This means that for any sub-block $\forall x^{i_0}$, we have

$$P\big(O^{i_0} = x^{i_0}\big) = q(x^{i_0}).$$

When $i = i_0 + 1$, by the algorithm, we have that either $\tau \geq i_0 + 1$, where $O^{(i_0+1)}$ is an accepted token, or $\tau \leq i_0$, where $O^{(i_0+1)}$ is sampled according to $q_{\text{new}}(\cdot \mid O^{i_0})$. We have

$$P\big(O^{i_0+1} = x^{i_0+1}\big)$$

$$= P\big(O^{i_0+1} = x^{i_0+1},\ \tau \geq i_0 + 1\big) + P\big(O^{i_0+1} = x^{i_0+1},\ \tau \leq i_0\big)$$

$$= P\big(O^{i_0+1} = x^{i_0+1},\ \tau \geq i_0 + 1\big) + P\big(O^{i_0} = x^{i_0},\ \tau \leq i_0\big) \cdot p_{\text{res}}\big(x^{(i_0+1)}\big)$$

$$= P\big(O^{i_0+1} = x^{i_0+1},\ \tau \geq i_0 + 1\big)$$

$$\quad + \Big(P\big(O^{i_0} = x^{i_0}\big) - P\big(O^{i_0} = x^{i_0},\ \tau \geq i_0 + 1\big)\Big) \cdot q_{\text{new}}\big(x^{(i_0+1)} \mid x^{i_0}\big)$$

$$= q(x^{i_0+1})\left(1 - \left(1 - \min\left\{\frac{p(x^{i_0+1})}{q(x^{i_0+1})}, 1\right\}\right)^K\right)$$

$$\quad + \Big(q(x^{i_0}) - \sum_x q(x^{i_0}, x)\left(1 - \left(1 - \min\left\{\frac{p(x^{i_0},x)}{q(x^{i_0},x)}, 1\right\}\right)^K\right)\Big) \cdot q_{\text{new}}\big(x^{(i_0+1)} \mid x^{i_0}\big)$$

$$= q(x^{i_0+1}),$$

where $q_{\text{new}}\big(x^{(i_0+1)} \mid x^{i_0}\big)$ is defined in Algorithm 2.

This completes the induction. Therefore, for all $i \leq L$, we have

$$P\big(O^i = x^i\big) = q(x^i).$$

□

## B PROOF OF THEOREM 5.3

First, we prove Lemma 5.1. For all couplings $\pi \in \Pi_{\text{CIC}}(p^{\otimes k}, q)$, let $(X^L, y^L) \sim \pi$. We further define a draft sequence $X_1^L$ extracted from $X^L$, and denote the accepted sequence as $Y$. Then, we have

$$\mathbb{E}[\tau] = \sum_{i=1}^{L} P(\tau \geq i)$$

$$= \sum_{\tau=1}^{L} \sum_{x^\tau} P(Y^\tau = x^\tau, X^L \text{ has } x^\tau)$$

$$= \sum_{\tau=1}^{L} \sum_{x^\tau} q(x^\tau) P(X^L \text{ has } x^\tau \mid Y^\tau = x^\tau)$$

$$= \sum_{\tau=1}^{L} \sum_{x^\tau} q(x^\tau) \Big[ 1 - \big(1 - P(X_1^\tau = x^\tau \mid Y^\tau = x^\tau)\big)^K \Big]$$

$$\leq \sum_{\tau=1}^{L} \sum_{x^\tau} q(x^\tau) \Big[ 1 - \big(1 - \min\big\{\frac{p(x^\tau)}{q(x^\tau)}, 1\big\}\big)^K \Big]. \tag{13}$$

In the last step of Eq. (13), we explicitly use the upper bound derived from the optimal transport principle:

$$P(X_1^\tau = x^\tau \mid Y^\tau = x^\tau) \leq \min\Big\{\frac{p(x^\tau)}{q(x^\tau)}, 1\Big\}.$$

which completes the proof.

Applying Lemma A.1 completes the proof of Theorem 5.3. In SpecTr-GBV, we have:

$$\mathbb{E}[\tau] = \sum_{i=1}^{L} P(\tau \geq i)$$

$$= \sum_{i=1}^{L} \sum_{x^i} P(X^L \text{ has } x^i, \tau^{x^i} \geq i)$$

$$= \sum_{\tau=1}^{L} \sum_{x^\tau} q(x^\tau) \left[ 1 - \left(1 - \min\Big\{\frac{p(x^\tau)}{q(x^\tau)}, 1\Big\}\right)^K \right].$$

## C ADDITIONAL EXPERIMENTAL RESULTS

In this section, we report the mean and standard deviation of each metric over 1,000 test prompts (draft length $L = 8$, temperature $T = 0.4$, draft number $K = 3$) across five datasets, using `CodeLlama-13B` with `CodeLlama-7B` as the draft model, and `Vicuna-13B` with `Vicuna-7B` as the draft model, as shown in Table 3. For both the `CodeLlama` and `Vicuna` models, we observe trends consistent with those on `DeepSeek`. In the `CodeLlama 13B-7B` setting, SpecTr-GBV outperforms SD, SpecTr, and GBV, achieving up to 8.0%, 0.9%, and 6.1% higher average BE, and 15.4%, 5.2%, and 17.4% higher average SR, respectively. In the `Vicuna 13B-7B` setting, SpecTr-GBV also outperforms all baselines, yielding up to 12.2%, 2.4%, and 8.9% higher average BE, along with 21.5%, 9.2%, and 21.5% higher average SR.

We also present ablation results using `DeepSeek-6.7B` as the target model and `DeepSeek-1.3B` as the draft model. Experiments are conducted on the HumanEval dataset with five random seeds, analyzing the effects of three key hyperparameters—$L$, $K$, and $T$—in the `6.7B-1.3B` setting.

**(1) Effect of draft length $L$:** We compare SpecTr-GBV in the `6.7B-1.3B` setting against baselines under varying draft lengths $L = 4, 8, 12, 16$, with temperature $T = 0.4$ and draft number $K = 3$.

Table 3: Performance comparison of SpecTr-GBV with baselines using `CodeLlama` and `Vicuna` models. The results show that SpecTr-GBV consistently outperforms all baselines across five datasets with both model families.

| Setting | Dataset | Metric | AR | SD | SpecTr | GBV | SpecTr-GBV |
|---|---|---|---|---|---|---|---|
| CodeLlama-13B CodeLlama-7B L=8 T=0.4 K=3 | HumanEval | BE | 1 | $7.58 \pm 0.76$ | $7.99 \pm 0.70$ | $7.59 \pm 0.79$ | $\mathbf{8.02 \pm 0.73}$ |
| | | SR | 1 | $1.45 \pm 0.15$ | $1.67 \pm 0.17$ | $1.42 \pm 0.15$ | $\mathbf{1.67 \pm 0.18}$ |
| | GSM8K | BE | 1 | $7.33 \pm 0.69$ | $7.89 \pm 0.51$ | $7.52 \pm 0.61$ | $\mathbf{7.95 \pm 0.53}$ |
| | | SR | 1 | $1.18 \pm 0.12$ | $1.48 \pm 0.24$ | $1.18 \pm 0.11$ | $\mathbf{1.51 \pm 0.22}$ |
| | MGSM | BE | 1 | $6.83 \pm 1.11$ | $7.36 \pm 0.95$ | $6.89 \pm 1.16$ | $\mathbf{7.39 \pm 0.96}$ |
| | | SR | 1 | $1.28 \pm 0.21$ | $1.39 \pm 0.18$ | $1.26 \pm 0.21$ | $\mathbf{1.51 \pm 0.21}$ |
| | LM1B | BE | 1 | $7.27 \pm 1.06$ | $7.81 \pm 0.81$ | $7.52 \pm 0.95$ | $\mathbf{7.88 \pm 0.82}$ |
| | | SR | 1 | $1.15 \pm 0.19$ | $1.17 \pm 0.18$ | $1.16 \pm 0.16$ | $\mathbf{1.28 \pm 0.23}$ |
| | Alpaca | BE | 1 | $7.08 \pm 0.94$ | $7.56 \pm 0.84$ | $7.20 \pm 0.93$ | $\mathbf{7.70 \pm 0.77}$ |
| | | SR | 1 | $1.07 \pm 0.14$ | $1.06 \pm 0.11$ | $1.05 \pm 0.13$ | $\mathbf{1.14 \pm 0.13}$ |
| | Average | BE | 1 | 7.21 | 7.72 | 7.34 | **7.79** |
| | | SR | 1 | 1.23 | 1.35 | 1.21 | **1.42** |
| Vicuna-13B Vicuna-7B L=8 T=0.4 K=3 | HumanEval | BE | 1 | $6.44 \pm 0.80$ | $6.88 \pm 0.78$ | $6.50 \pm 0.84$ | $\mathbf{6.96 \pm 0.74}$ |
| | | SR | 1 | $1.18 \pm 0.14$ | $1.20 \pm 0.13$ | $1.15 \pm 0.16$ | $\mathbf{1.31 \pm 0.14}$ |
| | GSM8K | BE | 1 | $5.96 \pm 0.69$ | $6.61 \pm 0.67$ | $6.13 \pm 0.72$ | $\mathbf{6.74 \pm 0.66}$ |
| | | SR | 1 | $1.16 \pm 0.14$ | $1.56 \pm 0.17$ | $1.15 \pm 0.14$ | $\mathbf{1.67 \pm 0.18}$ |
| | MGSM | BE | 1 | $5.94 \pm 1.09$ | $6.40 \pm 1.03$ | $6.11 \pm 1.03$ | $\mathbf{6.65 \pm 1.06}$ |
| | | SR | 1 | $1.09 \pm 0.23$ | $1.16 \pm 0.29$ | $1.09 \pm 0.20$ | $\mathbf{1.30 \pm 0.32}$ |
| | LM1B | BE | 1 | $5.04 \pm 1.30$ | $5.71 \pm 1.27$ | $5.30 \pm 1.32$ | $\mathbf{5.82 \pm 1.28}$ |
| | | SR | 1 | $0.91 \pm 0.23$ | $0.97 \pm 0.20$ | $0.92 \pm 0.23$ | $\mathbf{1.01 \pm 0.23}$ |
| | Alpaca | BE | 1 | $5.57 \pm 0.88$ | $6.17 \pm 0.90$ | $5.79 \pm 0.79$ | $\mathbf{6.32 \pm 0.83}$ |
| | | SR | 1 | $1.01 \pm 0.16$ | $1.04 \pm 0.14$ | $1.02 \pm 0.18$ | $\mathbf{1.21 \pm 0.22}$ |
| | Average | BE | 1 | 5.79 | 6.35 | 5.97 | **6.50** |
| | | SR | 1 | 1.07 | 1.19 | 1.07 | **1.30** |

Table 4: Ablation results of SpecTr-GBV under different draft lengths $L$ with $T = 0.4$, $K = 3$, in the `DeepSeek-6.7B-1.3B` setting.

| $L$ | AR | | SD | | SpecTr | | GBV | | SpecTr-GBV | |
|---|---|---|---|---|---|---|---|---|---|---|
| | BE | SR | BE | SR | BE | SR | BE | SR | BE | SR |
| 4 | 1 | 1 | 4.42 | 1.19 | 4.57 | 1.15 | 4.44 | 1.19 | **4.62** | **1.26** |
| 8 | 1 | 1 | 7.06 | 1.19 | 7.53 | 1.15 | 7.06 | 1.19 | **7.70** | **1.28** |
| 12 | 1 | 1 | 9.39 | 1.12 | 10.11 | 1.16 | 9.89 | 1.14 | **10.29** | **1.26** |
| 16 | 1 | 1 | 11.23 | 1.06 | 12.22 | 1.13 | 11.58 | 1.08 | **12.49** | **1.23** |

As shown in Table 4, for $L = 4$, SpecTr-GBV achieves relative improvements in the BE metric of 4.5%, 1.1%, and 4.1% over SD, SpecTr, and GBV, respectively. At $L = 16$, the corresponding improvements increase to 11.2%, 2.2%, and 7.9%. For the SR metric at $L = 4$, SpecTr-GBV achieves gains of 5.9%, 9.6%, and 5.9% compared to SD, SpecTr, and GBV, respectively. At $L = 16$, the gains grow to 16.0%, 8.8%, and 13.9%, highlighting that SpecTr-GBV exhibits greater advantages at longer draft lengths. Notably, as $L$ increases from 4 to 16, BE consistently improves, while SR experiences a slight decline, which aligns with our findings in Section 6.4.

**(2) Effect of draft number $K$:** As shown in Fig. 3a, we compare the acceptance rates of SpecTr-GBV and SpecTr in the `DeepSeek-6.7B-1.3B` setting under different numbers of draft sequences $K = 1, 3, 5, 7$, with draft lengths $L = 8$ and 12. The experimental results are consistent with those observed in the `DeepSeek-33B-1.3B` setting: as $K$ increases, the acceptance rates of both SpecTr and SpecTr-GBV improve, with SpecTr-GBV consistently outperforming SpecTr by relative margins of 0.78%, 1.75%, 2.62%, and 2.75% for $K = 1, 3, 5$, and 7, respectively. Moreover, we observe that the advantage of SpecTr-GBV becomes more pronounced as $K$ increases, indicating its superior scalability with respect to the number of draft sequences.

**(3) Effect of temperature $T$:** As shown in Fig. 3b, we evaluate the BE and SR of SpecTr-GBV in the `6.7B-1.3B` setting under different temperatures $T = 0.1, 0.4, 0.7$ with draft lengths $L = 8$

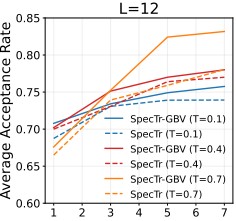 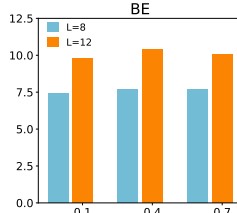 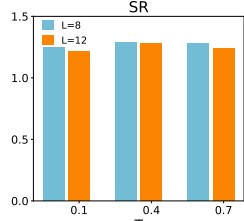

(a) Acceptance rate comparison between SpecTr-GBV and SpecTr with varying $K$ ($L = 8$ and 12) in the `DeepSeek-6.7B-1.3B` setting.

(b) BE and SR performance of SpecTr-GBV with varying temperature $T$ ($L = 8$ and 12, $K = 3$) in the `DeepSeek-6.7B-1.3B` setting.

Figure 3: Ablation results of SpecTr-GBV under different draft number (a) and temperature (b) in the `DeepSeek-6.7B-1.3B` setting.

and 12. Consistent with the `DeepSeek-33B-1.3B` setting, both BE and SR remain largely stable across different temperatures, further demonstrating the robustness of SpecTr-GBV to temperature variations.

# D ANALYSIS OF TIME EFFICIENCY: SPECTR-GBV VS. SPECTR

Table 5: Wall-clock time breakdown of SpecTr-GBV and SpecTr under `DeepSeek-33B-1.3B` and `DeepSeek-6.7B-1.3B` settings with T = 0.4, K = 3. We report the total wall-clock time (Total), number of decoding iterations for SD (Iter), time spent by the draft model (Draft) and target model (Target), verification algorithm overhead (Verification), other system overheads (e.g., cache operations) (Other), throughput (Tokens/s), and mean acceptance rate (Accept) for every data point. The $\delta$ row quantifies the percentage change of SpecTr-GBV relative to the SpecTr baseline.

| Setting | Dataset | Method | Total (s) | Iter | Draft (s) | Target (s) | Verification (s) | Other (s) | Tokens/s | Accept |
|---|---|---|---|---|---|---|---|---|---|---|
| DeepSeek-33B DeepSeek-1.3B $L$=12 | GSM8K | SpecTr | 19.24 | 73.46 | 11.97 | 6.05 | 1.01 | 0.20 | 27.48 | 0.521 |
| | | SpecTr-GBV | 18.03 | 70.13 | 11.57 | 5.79 | 0.47 | 0.20 | 29.64 | 0.555 |
| | | $\delta$ | -6.3% | -4.5% | -3.3% | -4.3% | -53.5% | -0.0% | +7.9% | +6.5% |
| | HumanEval | SpecTr | 12.91 | 51.72 | 8.91 | 2.83 | 1.03 | 0.12 | 40.99 | 0.772 |
| | | SpecTr-GBV | 11.39 | 50.92 | 8.24 | 2.75 | 0.26 | 0.12 | 46.79 | 0.788 |
| | | $\delta$ | -11.8% | -1.5% | -7.5% | -2.8% | -74.8% | -0.0% | +14.2% | +2.1% |
| DeepSeek-6.7B DeepSeek-1.3B $L$=8 | GSM8K | SpecTr | 12.36 | 79.27 | 8.71 | 2.44 | 1.00 | 0.19 | 42.01 | 0.695 |
| | | SpecTr-GBV | 11.12 | 78.07 | 8.20 | 2.39 | 0.35 | 0.19 | 46.93 | 0.711 |
| | | $\delta$ | -10.0% | -1.5% | -5.9% | -2.0% | -65.0% | -0.0% | +11.7% | +2.3% |
| | HumanEval | SpecTr | 10.09 | 69.71 | 7.39 | 1.56 | 1.01 | 0.12 | 51.85 | 0.816 |
| | | SpecTr-GBV | 9.05 | 67.65 | 7.17 | 1.52 | 0.25 | 0.12 | 57.67 | 0.838 |
| | | $\delta$ | -10.3% | -3.0% | -3.0% | -2.6% | -75.2% | -0.0% | +11.2% | +2.7% |

The results demonstrate that SpecTr-GBV achieves a higher acceptance rate, which in turn results in fewer iterations and reduced model processing time. Notably, as previously stated, SpecTr using K-SEQ requires a $O(|\Omega| \log(K))$ binary search for the $\rho$ parameter, while SpecTr-GBV computes acceptance probabilities directly at $O(|\Omega|)$. This theoretical efficiency is confirmed in practice as well, as the verification overhead for SpecTr-GBV is reduced by over $50\%$ compared to SpecTr in both settings.

# E THE USE OF LARGE LANGUAGE MODELS

Large Language Models (LLMs) were used solely to assist in the writing and polishing of this manuscript. Their application was limited to enhancing language and presentation, including correcting grammatical errors, rephrasing sentences for clarity and conciseness, and refining paragraph structure.

Importantly, the LLMs were not involved in the ideation, research methodology, experimental design, or data analysis. All research concepts, ideas, and analyses were conceived and conducted

exclusively by the human authors. The LLMs' contributions were restricted to improving the linguistic quality of the paper, with no influence on the scientific content.

The authors have thoroughly reviewed and edited all text and take full responsibility for the final content of the manuscript. We have ensured that the LLM-generated text adheres to ethical guidelines and does not contribute to plagiarism or scientific misconduct.

