# OpenReview forum: "SpecTr-GBV: Multi-Draft Block Verification Accelerating Speculative Decoding"
_ICLR.cc/2026/Conference — Submitted to ICLR 2026_

### Official Review · Reviewer_2szU · 2025-10-31

**Soundness:** 3
**Presentation:** 2
**Contribution:** 2
**Rating:** 2
**Confidence:** 3

**Summary:**

The paper proposes SpecTr-GBV, a speculative decoding (SD) method that combines multi-draft generation (à la SpecTr) with greedy block verification (GBV). The core idea is to formulate token verification as an optimal transport (OT) problem over entire draft blocks from multiple sequences, rather than position-by-position. The authors provide a theoretical proof that their method achieves the optimal expected number of accepted tokens for any fixed number of drafts, and this bound improves as the number of drafts increases. Extensive experiments across five datasets and four strong baselines demonstrate that SpecTr-GBV consistently outperforms prior art in both block efficiency and end-to-end speedup while preserving output quality.

**Strengths:**

1. Strong Theoretical Foundation: The paper provides a rigorous theoretical analysis, proving that SpecTr-GBV achieves the optimal expected acceptance length. This is a significant contribution that elevates the work beyond purely empirical improvements.
2. Comprehensive and Convincing Experiments: The empirical evaluation is thorough, covering multiple model families (DeepSeek, CodeLlama, Vicuna), diverse datasets (code, math, language modeling, instruction following), and extensive ablation studies on key hyperparameters (K, L, T).
3. Clear Practical Impact: The consistent speedup gains (e.g., +29.3% over SD, +8.2% over SpecTr in the 33B setting) demonstrate tangible benefits for real-world LLM inference.

**Weaknesses:**

1. The additional verification time cost caused by GBV+SpecTr needs to be theoretically analyzed and experimentally discussed.
2. The article is a combination of two methods in speculative decoding. This form of combination lacks significance.
3. All comparisons are against vanilla SD, SpecTr, and GBV—but miss recent strong contenders like Hydra, Medusa, or EAGLE, which use tree-structured or multi-head drafts. These often outperform i.i.d. multi-draft methods in practice. Without comparing to them, the claimed “state-of-the-art” status is unconvincing.

**Questions:**

1. Can you provide a breakdown of wall-clock time spent in each phase for SpecTr-GBV vs. SpecTr? It is an interesting question where the acceleration benefits come from.
2. Why not compare against tree-based speculative decoding methods? Your i.i.d. draft assumption may be fundamentally less efficient than structured candidate generation.

---

> ### Author Response · Authors · 2025-11-18
> **Rebuttal 1/2**
>
> We sincerely thank for your efforts in providing insightful comments and constructive feedback. In the following, we address the comments point by point.
> > Q1: All comparisons are against vanilla SD, SpecTr, and GBV—but miss recent strong contenders like Hydra, Medusa, or EAGLE, which use tree-structured or multi-head drafts. These often outperform i.i.d. multi-draft methods in practice. Without comparing to them, the claimed “state-of-the-art” status is unconvincing. Why not compare against tree-based speculative decoding methods?
>
> A:Thank you for your question. We want to clarify that our work does not aim to directly compete with tree-based speculative decoding methods on raw speedup benchmarks. Instead, our research pursues a distinct and equally vital objective: **to advance the theoretical foundations and practical performance within the Optimal Transport (OT) based framework for speculative decoding**.
>
> We acknowledge that tree-based methods (e.g., Medusa, EAGLE) have shown impressive speedup ratios (SRs) of 3.5x or higher. In contrast, previous OT-based works, including "Towards Optimal Multi-Draft Speculative" (ICLR 2025), "SpecTr: Fast Speculative Decoding via Optimal Transport" (NeurIPS 2023) and "Block Verification Accelerates Speculative Decoding" (ICLR 2025), have generally reported more modest SRs. However, this performance gap in terms of pure speed **does not diminish the unique and critical significance of the OT-based paradigm**.
>
> The core contribution of research in the OT framework lies in its dedicated focus on refining the verification strategy itself to enhance acceptance rates. This approach yields two outstanding and indispensable advantages:
> First, the OT framework provides a **clear and interpretable mechanism** for draft verification.
> Second, it allows for the provision of **solid theoretical guarantees and bounds**, a level of formal rigor often **less apparent in heuristic-driven, tree-based methods**.
>
> Furthermore, **tree-based approaches (like Medusa or EAGLE) frequently demand significant modifications to the model architecture or necessitate retraining**.  In scenarios with limited computational resources, these requirements can be prohibitive. The OT framework, in contrast, offers a more flexible and resource-efficient paradigm for speculative decoding.
>
> We chose the OT framework for its inherent strengths, and our work delivers substantially improved performance within this specific domain. We demonstrate a significant, theoretically-backed leap forward for OT-based research that outperforms its predecessors. For instance, on DeepSeek-33B-1.3B, our method achieves +2.3%  BE with +8.2%  SR over SpecTr, and +9.7% BE with +27.0% SR over GBV. Therefore, **our 'state-of-the-art' claim is firmly situated within the OT-based research lineage**.

---

> ### Author Response · Authors · 2025-11-18
> **Rebuttal 2/2**
>
> > Q2: The additional verification time cost caused by GBV+SpecTr needs to be theoretically analyzed and experimentally discussed. Can you provide a breakdown of wall-clock time spent in each phase for SpecTr-GBV vs. SpecTr? It is an interesting question where the acceleration benefits come from.
>
> A: Thank you for the insightful question regarding detailed time spent in each phase for SpecTr-GBV vs. SpecTr.
>
> We **present a wall-clock time breakdown** comparing SpecTr-GBV and SpecTr under the DeepSeek-33B-1.3B and DeepSeek-6.7B-1.3B settings, with $T=0.4$ and $K=3$ on dataset HumanEval and GSM8K **in Appendix D in the updated version**. We report the total wall-clock time (Total), number of decoding iterations for SD (Iter), time spent by the draft model (Draft) and target model (Target), verification algorithm overhead (Verification), other system overheads, e.g., cache operations (Other), throughput (Tokens/s), and mean acceptance rate (Accept) for every data point.
>
> The results demonstrate that SpecTr-GBV **achieves a higher acceptance rate**, which in turn results in fewer iterations and reduced model processing time. For instance, in the DeepSeek-33B-1.3B configuration on the GSM8K dataset, the mean acceptance rate and throughput (tokens/s) increased by 6.5% and 7.9%, respectively. This was accompanied by reductions in processing time of 3.3% for the draft model and 4.3% for the target model.
>
> Notably, SpecTr using K-SEQ requires a $O(|\Omega| \log(K))$ binary search for the $\rho$ parameter, while SpecTr-GBV computes acceptance probabilities directly at $O(|\Omega|)$, where $|\Omega|$ denotes the vocabulary size. **This theoretical efficiency is confirmed in practice as well**, as **the verification algorithm overhead for SpecTr-GBV is reduced** by over 50% compared to SpecTr in all scenarios. In the DeepSeek-33B-1.3B setting on the GSM8K dataset, the verification algorithm overhead for SpecTr is 1.01s per data point, compared to 0.47s for SpecTr-GBV, representing a reduction of 53.5%. This decrease also contributes to the overall acceleration of SpecTr-GBV, and we present theoretical analysis in the main text. The other system overheads remain consistent across all configurations.
>
> Therefore, the acceleration benefit of SpecTr-GBV over SpecTr stems directly from this more computationally efficient verification phase.
> > Q3: The article is a combination of two methods in speculative decoding. This form of combination lacks significance.
>
> A:We appreciate the feedback and understand the concern regarding the perceived novelty of our approach. However, we would like to emphasize that the significance of our work lies precisely in the principled fusion of multi-draft strategies and block verification—a challenge that, to our knowledge, has not been successfully addressed.
>
> Prior works have focused on one technique or the other, leaving a critical gap. Our contribution, SpecTr-GBV, **is not a straightforward combination but a carefully designed extension of block verification to the more complex multi-draft setting. This required us to rethink the verification process itself to maintain correctness and maximize efficiency across multiple candidate sequences.**
>
> The value of this fusion is substantiated by our results. We prove theoretically that SpecTr-GBV achieves a superior token acceptance rate. Our experiments further validate this, demonstrating substantial speedups and higher block efficiency while preserving output quality. While we respect that the definition of "novelty" can be subjective, we believe that successfully bridging these two lines of research to create a demonstrably superior method is a significant contribution. We hope the reviewer will consider the technical challenges we overcame and the strong results presented.

---

> ### Author Response · Authors · 2025-11-25
>
> Dear Reviewer 2szU,
>
> If we understand correctly, your main concerns are:
> (1) whether our comparisons sufficiently reflect relevant recent methods, and
> (2) how the wall-clock time of different decoding phases contributes to the speedup of SpecTr-GBV.
>
> In our response, we clarified the positioning of our work within the OT-based speculative decoding framework and explained why certain methods are not directly comparable in our setting. We also added detailed wall-clock time analysis showing that SpecTr-GBV achieves substantially lower verification overhead and improved overall efficiency.
>
> We kindly ask you to review our responses and let us know whether they address your concerns. Please feel free to reach out if anything remains unclear. We truly appreciate your time and feedback.

---

> ### Comment · Reviewer_2szU · 2025-11-26
>
> Thanks for the authors' responses. I acknowledge their claim of 'not a straight forward combination' and appreciate the speed related experiments in the appendix.
>
> However, I still have some doubts about OT based methods. Compared to tree-based methods, in industrial deployment, some modifications of models or some retraining are completely affordable costs. OT based methods still have certain disadvantages if their speed cannot completely match or surpass tree based methods. But as a research paper, I still support the study of OT based methods.
>
> Based on the above, I will increase my score from 2 to 4 and maintain it at a slightly negative rating.

---

> > ### Author Response · Authors · 2025-11-26
> >
> > Thank you very much for your thoughtful follow-up and for increasing your score. We sincerely appreciate your support for the study of OT-based speculative decoding.
> >
> > While, at the current stage, OT-based approaches may not yet achieve the same level of speedup as tree-based methods, we believe that—with further engineering optimizations and system-level improvements—the OT framework has strong potential to achieve both theoretical rigor and practical efficiency in real-world deployments.
> >
> > Your feedback is highly encouraging, and we are grateful for your balanced evaluation and constructive perspective.

---

### Official Review · Reviewer_yDb9 · 2025-11-01

**Soundness:** 1
**Presentation:** 3
**Contribution:** 2
**Rating:** 2
**Confidence:** 4

**Summary:**

The paper revisits speculative decoding (SD) by unifying multi-draft verification (SpecTr) with block-level verification (GBV). The method, SpecTr-GBV, formulates verification as an OT problem over draft/target blocks, gives closed-form acceptance/residual rules (Eqs. (2)–(4)), adds a distribution-modification step to preserve fidelity across iterations (Alg. 2, Thm. 3.2), and claims to achieve the maximum expected acceptance length for any fixed number of drafts K (Lemma 4.1,Thm. 4.3).

**Strengths:**

Re-tackles an important bottleneck. Accelerating multi‑draft verification is a natural and valuable direction.

Clear paper and algorithm presentation.

**Weaknesses:**

My main concern is theoretical soundness. Theory appears not rigorous to me.

1. The quantity $h_{ik}$ from Eq. (2) is used as a probability but can exceed 1 for $K=1$.

This paper assume $h_{ik}$ to be a probability. In page 5, it saids "Each token sub-block ... is accepted with probability $h_{ik}$". The same “(h) as a probability” usage happens in Appendix Eq. (11) in the proof. So for the proof to be valid, $h_{i,k}$ need to be no larger than 1.

However, setting K=1 in Eq. (2) yields

$$h\_{ik}=\dfrac{A+\Delta\_+}{A-\Delta},A=\sum\_x (q(x_i,x)-p(x_i,x))\_+,\Delta = q(x_i)-p(x_i)$$

If $\Delta>0$ (i.e., $q(x_i)>p(x_i)$), then $h_{ik}>1$.

Thus $h_{ik}$ is not a valid probability under the paper’s own acceptance test and proof steps.


2. Lemma 4.1 relies on a conditional‑independence step that is not guaranteed by the coupling.
See proof in Eq. (13).

**Questions:**

SpecTr optimality vs. K‑SEQ. My understanding is that SpecTr’s exact optimality follows from solving an OT, which is generally intractable, hence K‑SEQ is an approximation. In your method, Algorithm 1 does not solve an LP. In the (L=1) case, is SpecTr‑GBV guaranteed to match SpecTr’s optimal acceptance rate? If yes, why it could avoid LP which is necessary for SpecTr? If not, how does Thm. 4.3 still claim optimality?

---

> ### Author Response · Authors · 2025-11-18
> **Rebuttal**
>
> We sincerely thank for your efforts in providing insightful comments and constructive feedback. In the following, we address the comments point by point.
> >Q1: $h_{ik}$ from Eq.(2) is used as a probability but can exceed 1 for $K=1$.
>
> A:We apologize for the confusion. This is a typo in the equation for $h_{ik}$. We mistakenly used a "+" in the numerator instead of the correct "-". It is corrected in the updated version.
>
> The correct formulation is
> $$
> h_{ik}=
> \frac{
>     \sum_{x} q(x^{i}, x)
>     (1 - \min \{( \frac{p(x^{i}, x)}{q(x^{i}, x)},\ 1 )\} )^K - q(x^{i})
>     (1 - \min \{ (\frac{p(x^{i})}{q(x^{i})},\ 1 )\} )^K
> }{
>     1 - (1 - p(x^{i}))^K - q(x^{i}) + \sum_{x} q(x^{i}, x)
>     (1 - \min \{ (\frac{p(x^{i}, x)}{q(x^{i}, x)},\ 1) \} )^K
> }
> $$, which is indeed a valid probability.
>
> In K=1 scenario, we will prove that $h_{ik}$ is a valid probability by showing it is bounded by 0 and 1.
>
> $$h_{ik} = \frac{\sum_{x} (q(x^{i}, x) - \min\{(p(x^{i}, x), q(x^{i}, x))\}) - q(x^{i}) + \min\{(p(x^{i}), q(x^{i}))\}}{p(x^{i}) - q(x^{i}) + \sum_{x} (q(x^{i}, x) - \min\{(p(x^{i}, x), q(x^{i}, x))\})}$$
>
> $$=\frac{\min\{(p(x^{i}), q(x^{i}))\}-\sum_{x} \min\{(p(x^{i}, x), q(x^{i}, x))\} }{p(x^{i})-  \sum_{x}  \min\{(p(x^{i}, x), q(x^{i}, x))\}}$$
>
> First we prove the non-negativity that $h_{ik} \ge 0$ by showing both the numerator and denominator are non-negative.
> Since $$\sum_{x} \min\{(p(x^{i}, x), q(x^{i}, x))\} \le \min\{(\sum_{x} p(x^{i}, x), \sum_{x} q(x^{i}, x))\}=\min\{(p(x^{i}), q(x^{i}))\}$$, the numerator is non-negative.
> For denominator, since $$\sum_{x} \min\{(p(x^{i}, x), q(x^{i}, x))\} \le \sum_{x} p(x^{i}, x) =p(x^{i}),$$ the denominator is also non-negative.
> Therefore, $h_{ik} \ge 0$.
>
> Then we show that $h_{ik} \le 1$.
> Compare the numerator and the denominator.
> For numerator: $\min\{(p(x^{i}), q(x^{i}))\}-\sum_{x} \min\{(p(x^{i}, x), q(x^{i}, x))\}$.
> For denominator: $p(x^{i})- \sum_{x} \min\{(p(x^{i}, x), q(x^{i}, x))\}$.
> As $\min\{(p(x^{i}), q(x^{i}))\} \le p(x^{i})$, the numerator is always less than or equal to the denominator.
> Therefore, $h_{ik} \le 1$.
>
> This confirms that $h_{ik}$ is a valid probability. We thank you again for your invaluable feedback.
>
> >Q2: Lemma 4.1 relies on a conditional‑independence step that is not guaranteed by the coupling. See proof in Eq.(13).
>
> A: We appreciate the technical query regarding Eq. (13). **The conditional independence is indeed guaranteed**, as it stems directly from the autoregressive model's inference process in the multi-draft setting. With the batch size of $K$, the draft model ($\mathcal{M}_s$) generates $K$ draft sequences **i.i.d. in parallel**. The coupling $\pi$ is a transport plan constructed after these i.i.d. drafts have been generated. It defines a joint distribution between the entire set of i.i.d. draft sequences (distributed as $p^{\oplus K}$) and the target sequence distribution $q$.
>
> So this coupling **does not "destroy" the inherent i.i.d. nature of the draft proposals**. The step in Eq. (13) leverages this exact i.i.d. property to calculate the probability of at least one of the $K$ independent drafts matching the target.
>
> > Q3: SpecTr optimality vs. K‑SEQ. My understanding is that SpecTr’s exact optimality follows from solving an OT, which is generally intractable, hence K‑SEQ is an approximation. In your method, Algorithm 1 does not solve an LP. In the (L=1) case, is SpecTr‑GBV guaranteed to match SpecTr’s optimal acceptance rate? If yes, why it could avoid LP which is necessary for SpecTr? If not, how does Thm. 4.3 still claim optimality?
>
> A: Thank you for this question. For any given $L$ (including $L=1$), SpecTr-GBV is guaranteed to achieve the theoretical optimal acceptance rate, which SpecTr's K-SEQ algorithm only approximates. We avoid the LP because **the optimality of an OT problem can be established in two ways**:
>
> (1) Solve the LP: Run a solver to find the optimal coupling $\pi$, which is precisely what the SpecTr algorithm does.
>
> (2) Construct a plan: Design a specific algorithm (a set of acceptance/residual probabilities) and prove that this algorithm's expected acceptance exactly matches the theoretical maximum bound.
>
> **The GBV paper followed the second method, and our SpecTr-GBV does as well**. We first derive the theoretical upper bound on the expected acceptance length for any batch size $K$ (Thm. 5.3, which is Thm. 4.3 in previous version). Then, we designed our algorithm's probabilities ($h_{ik}$, $h_{Lk}$, and $p_{res}$ in Eqs.2, 3, and 4) to be the exact values required to meet this bound while satisfying all OT probability constraints.

---

> ### Author Response · Authors · 2025-11-25
>
> Dear Reviewer yDb9,
>
> If we understand correctly, your main concern lies in the theoretical soundness of our work. In our response, we have clarified that the issue was caused by a typographical error involving a plus/minus sign, which has now been corrected in the revised manuscript.
>
> We kindly ask you to take a moment to review our response and let us know whether it satisfactorily addresses your concern. Please do not hesitate to let us know if anything remains unclear. We sincerely appreciate the opportunity to clarify our work and would be very grateful for any additional feedback you may have.
>
> Thank you very much for your time and consideration.

---

> ### Comment · Reviewer_yDb9 · 2025-11-27
> **Thank you for the rebuttal: some issues fixed (score raised), but some concerns remain**
>
> Thank you for the revision.
>
> First, I am glad to see that the obvious issue in Eq.(2), that $h$ could exceed $1$ due to an incorrect $+$ sign, has been fixed. This correction fix one counterexample I previously raised, improving the soundness of this paper, so I will accordingly raised my score to 4 (somehow I cannot edit my previous review in the system now).
>
> However, I do not believe that the theoretical soundness concerns have been fully resolved.
> In particular, the derivation in Lemma 5.1 still relies on the conditional-independence step in
> Eq. (13), which assumes
> $$
> P\left(\bigcup_k\\{X\_k^\tau = x^\tau\\} \mid Y^\tau=x^\tau\right)
> = 1 - \bigl(1 - P(X\_1^\tau = x^\tau \mid Y^\tau = x^\tau)\bigr)^K,
> $$
> an equality that requires the $\\{X\_k^\tau\\}\_{k=1}^K$ to remain independent **given** $Y^\tau$.
> For a general coupling $\pi\in\Pi(p^{\otimes K},q)$, this conditional independence is not implied by
> unconditional i.i.d. sampling of drafts, conditioning on $Y_\tau$ can introduce arbitrary
> dependencies among the drafts. Therefore, the key step in Eq. (13) does not yet appear to be
> justified, and the resulting upper bound in Lemma 5.1, which Theorem 5.3 depends on, remains
> unproven as stated.
>
> If the authors have or develop a corrected argument establishing this bound without relying on
> conditional independence, or under clearly stated additional structural constraints on the coupling,
> I would be very happy to re-evaluate the analysis. I appreciate the efforts made so far and look
> forward to any further clarification.

---

> ### Author Response · Authors · 2025-11-29
> **Rebuttal 2: Clarification on Lemma 5.1**
>
> We sincerely thank the reviewer for the rigorous examination and for raising the score. We apologize for the confusion caused by the lack of explicit structural constraints in our initial derivation, and we fully appreciate your concern regarding Lemma 5.1 under general settings.
>
> We agree that within the mathematical space of *general couplings* $\pi \in \Pi(p^{\otimes K}, q)$, the draft tokens $\{X_k^\tau\}$ are not necessarily independent conditioned on $Y^\tau$. In such a general space, arbitrary dependencies (including negative correlations) can exist, which would indeed violate the independence assumption and potentially exceed the upper bound in Eq. (13).
>
> However, we would like to clarify that **our original intention was to construct an Optimal Transport formulation that specifically satisfies the property of a Conditionally Independent Coupling (CIC).**
>
> The rationale for this choice is fundamental: **the theoretical limit of CIC is physically attainable within the Speculative Decoding (SD) framework, whereas the theoretical optimum of General OT cannot.**
>
> Our work addresses SD, which fundamentally relies on *i.i.d.* draft generation. Moreover, practical verification schemes (like ours and K-SEQ) typically employ sequential processing, **which enforces independence conditioned on the verified token. This process intrinsically adheres to the CIC framework. Since any physically realizable verification scheme is bound by this structure, the ''optimal'' coupling is naturally constrained to the set where drafts remain independent given the target.**
>
> To address your concern and ensure theoretical rigor, we explicitly state this constraint in the revised submission (see Definition 4.1). We define the feasible set of couplings as $\Pi_{CIC} \subset \Pi(p^{\otimes K}, q)$, where for any $\pi \in \Pi_{CIC}$:
> $$ P_\pi(X_1, \dots, X_K \mid Y) = \prod_{k=1}^K P_\pi(X_k \mid Y). $$
> Under this structural constraint—which matches our intended design scope, Eq. (13) strictly holds.
>
> This clarification not only fixes the derivation but also highlights a key distinction from previous work (e.g., SpecTr [NeurIPS 2023]):
>
> (1)**General OT (SpecTr):** By considering general couplings, SpecTr derived a "loose" upper bound. Crucially, such a bound derived from general OT is **physically unattainable** for verification algorithms like K-SEQ that process drafts sequentially. This explains **why their algorithm can only approximate that loose bound** (specifically, within a factor of $1 - 1/e$).
>
> (2)**CIC OT (Ours):** By focusing on the intended CIC regime, we derive a *tight upper bound* (Eq. 5) that reflects the true theoretical limit of i.i.d. speculative decoding. Since the CIC limit is **strictly attainable** in this framework, Theorem 5.3 proves that SpecTr-GBV **exactly achieves** this tight bound.
>
> In the final version, we revised the statement of Definition 4.1 to explicitly restrict the optimization to the set of Conditionally Independent Couplings ($\Pi_{CIC}$). This revision aligns the mathematical formulation with our design intention and the physical reality of the problem, confirming that our method strictly achieves the derived upper bound.
>
> We once again thank you for your invaluable feedback.

---

### Official Review · Reviewer_6bFy · 2025-11-04

**Soundness:** 3
**Presentation:** 3
**Contribution:** 3
**Rating:** 6
**Confidence:** 3

**Summary:**

This paper proposes SpecTr-GBV, a speculative decoding (SD) algorithm that unifies multi-draft decoding and greedy block verification (GBV) under a single framework. It formulates the verification process as an optimal transport (OT) problem between multiple draft token blocks and the target token block, and proves that this formulation achieves the optimal expected acceptance length for any fixed number of drafts. The method theoretically extends prior works—SpecTr (Sun et al., 2023) and GBV (Sun et al., 2024b)—and empirically shows consistent speed-ups across five datasets (HumanEval, GSM8K, MGSM, LM1B, Alpaca) using various target/draft model pairs (DeepSeek-33B/1.3B, 6.7B/1.3B, etc.).

**Strengths:**

The paper is very clearly written; The paper unifies two influential lines of speculative decoding: multi-draft optimal transport (SpecTr) and block-level greedy verification (GBV). The evaluation is very comprehensive involving multiple models. The paper also provides in-depth analysis in terms of temperature, draft lengths.

**Weaknesses:**

The paper is technically correct and experimentally comprehensive, but the overall contribution is incremental. It mainly combines two existing speculative decoding mechanisms (SpecTr and GBV) under one unified framework, without introducing new conceptual insights or architectural innovation.

**Questions:**

N/A

---

> ### Author Response · Authors · 2025-11-18
> **Rebuttal**
>
> >Q: The paper is technically correct and experimentally comprehensive, but the overall contribution is incremental.
>
> A: We thank the reviewer for this question. Prior work in speculative decoding primarily adopts separate acceleration mechanisms, either through multi-draft strategies or through block verification. As far as we know, the combination of these two powerful techniques has remained unexplored. Our proposed method **uniquely** bridges this methodological gap by designing a new block-verification strategy in multi-draft setting, and we provide theoretical proof that SpecTr-GBV achieves a superior token acceptance rate compared to current existing methods, and practically achieves superior speedup and significantly higher block efficiency while preserving output quality.

---

### Official Review · Reviewer_mAif · 2025-11-11

**Soundness:** 3
**Presentation:** 3
**Contribution:** 3
**Rating:** 8
**Confidence:** 2

**Summary:**

The paper introduces a new method for speculative decoding, SpecTr-GBV. The paper conducts a fairly thorough theoretical analysis of their method and prove that it 1) faithfully samples from the desired distribution and 2) achieves expected acceptance length that is (in some sense, compared to a class of methods) optimal. The paper then presents several sets of experiments demonstrating the advantage of their method compared to other methods.

**Strengths:**

I believe the paper has an interesting and compelling mix of theory and empirics. It is particularly compelling that the proposed method SpecTr-GBV achieves the upper bound for expected acceptance length, and moreover that this performance gain is reflected in experiments. Specifically there is a major performance gain in Block Efficiency (BE), which (if my understanding is correct) corresponds to improvement in $\mathbb{E}[\tau]$ as suggested by the theoretical guarantee Theorem 4.3. The theoretical bound also subsumes that of GBV when $K=1$, which makes the extension to larger $K$ and the general approach convincing.

Overall, I think the method makes sense, has relevant theoretical guarantees, and has practical significance. The paper is also overall well written and I found it easy to follow.

**Weaknesses:**

I am relatively unfamiliar with the prior literature on speculative decoding, my background is on the theory side, in high dimensional sampling algorithms. As such it is not clear to me how novel the proposed algorithm is, and how much of a contribution it represents vs the literature. It seems like it is built off of similar ideas as KSEQ and GBV, albeit with several modifications / combining ideas (this is even stated explicitly by the authors in line 197 at start of subsection 3.1). That is fine, but it's unclear to me how much of a development this paper comprises in the literature.

Another weakness is about theoretical guarantees of efficiency (some of these concerns apply also to previously developed algorithms). The method requires generating K draft sequences, does this make the method slower (in theory) than GBV? Also eq (2) requires summing over all possible tokens x, similarly (4) requires sampling from a distribution over tokens x with no formal guarantees of `niceness' of a discrete univariate distribution such as unimodality (note that these issues also apply to methods like GBV). Does the proposed method require many more such inefficient computations than previously developed methods? Such concerns are certainly of practical importance as well, e.g. as noted in lines 418-422 that the generation of the token from the draft model cannot be ignored in practice.

Finally, a few writing suggestions: section 6 on related work could be moved earlier. Also Algorithm 1 is long and very hard to read.

**Questions:**

-Could the authors please provide some intuition on how the attained bound in Theorem 4.3 seems to behave in terms of L, K? Of course this depends on p and q but some idea on how this might behave (e.g. linear in L, $\sqrt{L}$, dependence on $K$ etc for some particular but representative cases of p, q) would be much appreciated. Some properties (Monotonicity, Convergence, Consistency) are written but a better idea of the quantitative behavior would be useful.

-Is there an idea of the theoretical guarantees for SpecTr-GBV's computational efficiency vs pre-existing methods in the literature?

-Theorem 4.3 shows that SpecTr-GBV satisfies the upper bound in Lemma 4.1, i.e. Lemma 4.1 is tight for SpecTr-GBV. Some intuition on why this is the case would be great, could the authors please provide some?

---

> ### Author Response · Authors · 2025-11-18
> **Rebuttal 1/2**
>
> We sincerely thank for your efforts in providing insightful comments and constructive feedback. In the following, we address the comments point by point.
> > Q1: It's unclear to me how much of a development this paper comprises in the literature.
>
> A: Prior work in speculative decoding primarily adopts separate acceleration mechanisms, either through multi-draft strategies or through block verification. As far as we know, the combination of these two powerful techniques has remained unexplored. Our proposed method **uniquely** bridges this methodological gap by designing a new block-verification strategy in multi-draft setting, and we provide theoretical proof that SpecTr-GBV achieves a superior token acceptance rate compared to current existing methods, and practically achieves superior speedup and significantly higher block efficiency while preserving output quality.
>
> > Q2: Section 6 on related work could be moved earlier.
>
> A: Thank you for this excellent suggestion. The Related Work section has been moved to Section 2 for better contextual flow.
>
> > Q3: The method requires generating K draft sequences. Does this make the method slower (in theory) than GBV?
>
> A: During the inference of the autoregressive model, with batch size as $K$, the draft model ($\mathcal{M}_s$) generates $K$ draft sequences i.i.d. **in parallel**. Consequently, the time required for this parallel draft generation is equivalent to the time needed to generate a single sequence of equivalent length.
>
> Given $K$ distinct context sequences $x_1^L, x_2^L, \dots, x_K^L$, the target model ($\mathcal{M}_b$) can compute the required conditional probabilities ($\mathcal{M}_b(\cdot |x_j^t)$) for all sequences and their proposed tokens simultaneously within a single, batched forward pass.
>
> In theory, the wall-clock time for this parallel verification step is comparable to that of processing a single sequence (i.e., a batch size of 1). Therefore, this multi-draft setting effectively parallelizes both the drafting and verification stages, enabling $K$ sequences to be processed in approximately **the same amount of time as one sequence**.
>
> > Q4: Also eq (2) requires summing over all possible tokens x. Is there an idea of the theoretical guarantees for SpecTr-GBV's computational efficiency vs pre-existing methods in the literature?
>
> A: We appreciate the concerns regarding implementation complexity. However, we find that the theoretical demands, such as summing over the vocabulary ($\Omega$) and performing sampling, do not translate into significant practical hurdles. To substantiate this, we present the following theoretical complexity analysis and also provide the analysis in the main text.
>
> For a single token selection step at a given position $x^{(i)}$, SpecTr employs K-SEQ and has a computational time complexity of $O(|\Omega| \log(K))$, where $|\Omega|$ denotes the vocabulary size. This cost is driven by the binary search K-SEQ uses to compute the optimal $\rho$ parameter, which adjusts the acceptance probability.
>
> In contrast, the token selection step for $x^{i}$ in both GBV and SpecTr-GBV has **a more favorable time complexity of $O(|\Omega|)$**. This cost is dominated by the need to calculate acceptance probabilities (e.g., $h_{ik}$), a process that requires a single sum over the entire vocabulary $\Omega$.
>
> This theoretical analysis—where $O(|\Omega|)$ is superior to $O(|\Omega| \log(K))$—is **consistent with our empirical results**. We present a wall-clock time breakdown comparison in Appendix D, and it is confirmed in practice that SpecTr-GBV exhibits lower verification times than SpecTr.
>
> > Q5: Could the authors please provide some intuition on how the attained bound in Theorem 4.3 seems to behave in terms of L, K?
>
> A: For K, as we stated in Remark 5.2（Remark 4.2 in previous version）, the upper bound converges to the maximum acceptance length $L$ as the number of drafts $K$ tends to infinity.
>
> As for L, in the ideal case, where the draft model $p$ closely approximates the target $q$, the ratio $p(x)/q(x) \approx 1$. This implies the expected acceptance length grows **linearly** with the draft length $L$. In realistic scenarios, $p$ does not perfectly match $q$. As $L$ increases, it becomes harder to generate long, consistently accepted sequences, resulting in sub-linear growth.

---

> ### Author Response · Authors · 2025-11-18
> **Rebuttal 2/2**
>
> > Q6: Theorem 4.3 shows that SpecTr-GBV satisfies the upper bound in Lemma 4.1, i.e. Lemma 4.1 is tight for SpecTr-GBV. Some intuition on why this is the case would be great, could the authors please provide some?
>
> A: Thank you for the question.
> The optimality of an OT problem can be **established in two ways**:
> (1) one may run a solver to obtain the optimal coupling $\pi$; or
> (2) one may design a concrete algorithm (i.e., a set of acceptance and residual probabilities) and prove that its expected acceptance exactly achieves the theoretical upper bound.
>
> **The GBV paper followed the second method, and our SpecTr-GBV does as well.** We first derive the theoretical upper bound on the expected acceptance length for any batch size $K$ (Thm. 5.3，which is Thm. 4.3 in previous version). Then, we designed our algorithm's probabilities ($h_{ik}$, $h_{Lk}$, and $p_{res}$ in Eqs. 2, 3, and 4) to be the exact values required to meet this bound while satisfying all OT probability constraints, thereby perfectly matching the optimal scenario described by the upper bound in Lemma 5.1 (Lemma 4.1 in previous version).

---

### Meta-Review · Area_Chair_3Jda · 2025-12-20

**Summary:**

**Paper summary.** This paper proposes SpecTr-GBV, combining multi-draft speculative decoding (SpecTr/OT-style) with greedy block verification (GBV-style). The paper’s headline claim is that the method is (i) lossless (samples from the correct target distribution) and (ii) achieves an optimal expected acceptance length for a given number of drafts, backed by theoretical derivations. The authors also report broad experiments across multiple model families and tasks.

**What happened in the discussion.** The reviews were polarized: two reviewers were strongly positive (6 and 8) and emphasized the theory + thorough experiments; two reviewers gave low scores (2 and 2) and raised concerns about novelty/baselines and, crucially, correctness/rigor of parts of the theory. The discussion centered on concrete theoretical issues: one reviewer pointed out an equation where a term used as a probability could exceed 1, and the authors acknowledged this was a sign error and corrected it. After that fix, the same reviewer still argued that a key bound relies on a conditional-independence step that does not hold for a general coupling, so the stated upper bound is not proven “as written” unless extra structural constraints are made explicit. The authors responded that their intended scope is a restricted coupling class (conditionally independent coupling) that matches i.i.d. draft generation in physically realizable speculative decoding, and they proposed to revise definitions/claims accordingly. In parallel, another reviewer questioned missing comparisons to strong recent speculative decoding methods (tree-based / multi-head drafts) and asked for wall-time breakdown to show the real source of speedup; authors added some timing analysis and clarified their intended positioning as OT-based research rather than competing with all tree-based methods.

**My assessment as AC.** The paper is ambitious and has a lot of work in it. However, the main selling point is a theoretical “optimality” statement. Right now, the forum record shows (a) at least one concrete derivation bug was present and fixed during rebuttal, and (b) there is still disagreement about whether the main bound is correct under the originally stated assumptions. Even if the authors’ restricted-scope argument is reasonable, the current state creates too much risk that the core theorem will be viewed as unclear or overstated by the broader community. Given how competitive ICLR is, I do not think we should accept a paper whose key theoretical claim is still contested in the discussion.

**Decision.** Reject. This is not saying the paper is not useful. The experiments and the direction are interesting, and the authors did engage in technical discussion. I recommend tightening the formal problem statement and theorem scope (state assumptions plainly, align definitions with what SD can actually realize), and adding clearer comparisons/time breakdown. With those changes, this could be a strong submission to a future venue.

**Reviewer Concerns:**

- **Reviewer yDb9 (rating 2, confidence 4)**: Raised the most serious theoretical concerns.
  - Point 1: A quantity used as an acceptance probability could exceed 1 (Eq. 2 as originally written). Authors acknowledged a sign typo and corrected it, and provided an argument that it is bounded in K=1.
  - Point 2: The upper-bound proof relies on a conditional-independence step that is not implied by a general OT coupling. Authors later clarified they intend to restrict the feasible set to “conditionally independent couplings (CIC)” that match i.i.d. draft generation, and they plan to update the definition/theorem scope accordingly. The reviewer still stated the bound is unproven “as written” unless these constraints are explicit.
  - Status: **not fully resolved** (scope and correctness of the main bound remain contested).
- **Reviewer 2szU (rating 2, confidence 3)**: Focused on novelty/baselines and system costs.
  - Point 1: Missing comparisons to strong recent speculative decoding baselines (Hydra/Medusa/EAGLE, tree-based methods), making “SOTA” claims hard to accept.
  - Point 2: Need a wall-time phase breakdown to show where speedup comes from and whether GBV+SpecTr increases verification overhead.
  - Authors replied with positioning (“OT-based framework focus”) and added timing breakdown experiments in the appendix; the reviewer stated they would raise the score to 4 but remain slightly negative.
  - Status: **partially resolved** (some evidence added, but baseline gap remains).
- **Reviewer 6bFy (rating 6, confidence 3)**: Found the paper clear and technically sound but viewed it as incremental (“unifies two existing mechanisms without new conceptual insight”). Authors mainly argued that the unified OT framing and proofs are the contribution. **Status:** partially resolved (this is a matter of taste).
- **Reviewer mAif (rating 8, confidence 2)**: Very positive; main request was intuition/interpretation of the theoretical bound’s dependence on L and K and novelty positioning. Authors provided explanation in rebuttal. **Status:** mostly resolved.

**Reviewer Scores:**

- **Reviewer mAif (rating 8, confidence 2)**: Very positive but low confidence; likely unchanged.
- **Reviewer 6bFy (rating 6, confidence 3)**: Positive but views contribution as incremental; likely unchanged.
- **Reviewer 2szU (rating 2, confidence 3)**: Negative mainly on novelty/baselines and overhead; after author responses and added timing experiments, they stated they would increase the score from 2 to 4 but remain slightly negative.
- **Reviewer yDb9 (rating 2, confidence 4)**: Negative on theoretical soundness; after the sign fix they stated they would raise the score to 4, but explicitly said the key upper bound still appears unproven as stated without additional structural constraints.

---

### Decision · Program_Chairs · 2026-01-26

Reject